# T cell self-reactivity during thymic development dictates the timing of positive selection

Lydia K Lutes[1], Zoë Steier[2], Laura L McIntyre[1], Shraddha Pandey[1], James Kaminski[3†], Ashley R Hoover[1‡], Silvia Ariotti[1§], Aaron Streets[2,3,4], Nir Yosef[2,3,4,5,6], Ellen A Robey[1]*

[1]Division of Immunology and Pathogenesis, Department of Molecular and Cell Biology, University of California, Berkeley, Berkeley, United States; [2]Department of Bioengineering, University of California, Berkeley, Berkeley, United States; [3]Center for Computational Biology, University of California, Berkeley, Berkeley, United States; [4]Chan Zuckerberg Biohub, San Francisco, United States; [5]Department of Electrical Engineering and Computer Sciences, University of California, Berkeley, Berkeley, United States; [6]Ragon Institute of MGH, MIT and Harvard, Cambridge, United States

*For correspondence:
erobey@berkeley.edu

Present address: †Division of Pediatric Hematology/Oncology, Boston Children's Hospital; Department of Medical Oncology, Dana Farber Cancer Institute, and Harvard Medical School, Boston, United States; ‡Oklahoma Medical Research Foundation, Oklahoma, United States; §Instituto de Medicina Molecular, João Lobo Atunes, Faculdade de Medicina da Universidade de Lisboa, Av. Professor Egas Moniz, Lisbon, Portugal

Competing interests: The authors declare that no competing interests exist.

**Abstract** Functional tuning of T cells based on their degree of self-reactivity is established during positive selection in the thymus, although how positive selection differs for thymocytes with relatively low versus high self-reactivity is unclear. In addition, preselection thymocytes are highly sensitive to low-affinity ligands, but the mechanism underlying their enhanced T cell receptor (TCR) sensitivity is not fully understood. Here we show that murine thymocytes with low self-reactivity experience briefer TCR signals and complete positive selection more slowly than those with high self-reactivity. Additionally, we provide evidence that cells with low self-reactivity retain a preselection gene expression signature as they mature, including genes previously implicated in modulating TCR sensitivity and a novel group of ion channel genes. Our results imply that thymocytes with low self-reactivity downregulate TCR sensitivity more slowly during positive selection, and associate membrane ion channel expression with thymocyte self-reactivity and progress through positive selection.

## Introduction

T cell fate is dictated by the strength of the interaction between the T cell receptor (TCR) and self-peptide major histocompatibility complexes (self-pMHCs) presented on thymic-resident antigen-presenting cells (APCs). While thymocytes whose TCRs interact too weakly or too strongly with self-pMHC undergo death by neglect or negative selection, respectively, thymocytes whose TCRs react moderately with self-pMHC can survive and give rise to mature CD4+ or CD8+ single-positive (SP) T cells. Naïve CD4 and CD8 T cells that emerge from the thymus were initially thought to be relatively homogeneous, with differences emerging only after antigenic stimulation. However, recent studies have shown that positively selected T cells span a range of self-reactivities, and that positive selection results in functional tuning of T cells based on their degree of self-reactivity (*Azzam et al., 1998*; *Fulton et al., 2015*; *Persaud et al., 2014*; *Weber et al., 2012*; *Mandl et al., 2013*). On naïve T cells, surface expression levels of the glycoprotein CD5 serve as a reliable marker for self-reactivity (*Azzam et al., 2001*; *Azzam et al., 1998*; *Hogquist and Jameson, 2014*). CD5 also negatively regulates TCR signals, thereby serving as part of the tuning apparatus (*Azzam et al., 2001*; *Azzam et al., 1998*; *Hogquist and Jameson, 2014*; *Persaud et al., 2014*; *Tarakhovsky et al.,*

*1995*). T cells with relatively high self-reactivity (CD5^high) expand more rapidly upon priming and exhibit greater sensitivity to cytokines (*Fulton et al., 2015*; *Persaud et al., 2014*; *Weber et al., 2012*; *Gascoigne and Palmer, 2011*), whereas T cells with relatively low self-reactivity (CD5^low) survive better in the absence of cytokines and exhibit stronger proximal TCR signals upon stimulation (*Persaud et al., 2014*; *Palmer et al., 2011*; *Smith et al., 2001*; *Cho et al., 2016*). How these distinct functional properties are imprinted during positive selection in the thymus remains unknown.

Positive selection is a multi-step process in which CD4+CD8+ double-positive (DP) thymocytes experience TCR signals, migrate from the thymic cortex to the medulla, and eventually downregulate either CD4 or CD8 co-receptor. During this process, thymocytes remain motile and experience serial, transient TCR signaling events, to eventually reach a threshold of signaling in order to complete positive selection (*Au-Yeung et al., 2014*; *Ross et al., 2014*; *Melichar et al., 2013*). Thymocytes bearing MHC-II specific TCRs take 1–2 days to complete positive selection and become mature CD4 SP thymocytes, whereas thymocytes bearing MHC-I specific TCRs give rise to CD8 SP thymocytes about 2–4 days after the initiation of positive selection (*Saini et al., 2010*; *Lucas et al., 1993*; *Kurd and Robey, 2016*). Interestingly, some studies report that CD8 T cells continue to emerge more than 1 week after the beginning of positive selection (*Saini et al., 2010*; *Lucas et al., 1993*). It remains unknown why some thymocytes move expeditiously through positive selection, while others require more time to mature.

TCR signaling leads to a rise in cytosolic calcium concentration, and calcium flux during positive selection induces a migratory pause that may prolong the interaction between thymocyte and APC, thus promoting positive selection (*Bhakta and Lewis, 2005*; *Bhakta et al., 2005*; *Melichar et al., 2013*). Calcium flux in thymocytes is also important for activation of the calcium-dependent phosphatase calcineurin and the downstream transcription factor NFAT, both of which are required for normal positive selection (*Gallo et al., 2007*; *Canté-Barrett et al., 2007*; *Macian, 2005*; *Neilson et al., 2004*; *Hernandez et al., 2010*; *Oh-hora and Rao, 2008*). Although the initial calcium flux following TCR stimulation comes from the release of calcium from the endoplasmic reticulum (ER) stores, calcium influx from the extracellular space is needed for prolonged calcium signals (*Oh-hora and Rao, 2008*). During mature T cell activation, depletion of ER calcium stores triggers the influx of calcium via calcium release-activated calcium (CRAC) channels. However, components of the CRAC channels are not required for positive selection (*Oh-hora, 2009*; *Gwack et al., 2008*; *Vig and Kinet, 2009*), and the mechanisms that maintain intracellular calcium levels during positive selection remain unknown.

Just after completing TCRαβ gene rearrangements and prior to positive selection, DP thymocytes (termed preselection DP) are highly sensitive to low-affinity ligands (*Davey et al., 1998*; *Lucas et al., 1999*; *Gaud et al., 2018*). This property is thought to enhance the early phases of positive selection, but must be downregulated to prevent mature T cells from responding inappropriately to self (*Hogquist and Jameson, 2014*). A number of molecules have been identified, which are involved in modulating TCR signaling in DP thymocytes and which are downregulated during positive selection. These include the microRNA miR-181a, which enhances TCR signaling by downregulating a set of protein phosphatases (*Li et al., 2007*); Tespa1, a protein that enhances calcium release from the ER (*Lyu et al., 2019*; *Liang et al., 2017*); components of a voltage-gated sodium channel (VGSC), which prolongs calcium flux via an unknown mechanism (*Lo et al., 2012*); and Themis, a TCR-associated protein with controversial function (*Kakugawa et al., 2009*; *Patrick et al., 2009*; *Lesourne et al., 2009*; *Johnson et al., 2009*; *Fu et al., 2009*; *Choi et al., 2017a*; *Fu et al., 2013*; *Choi et al., 2017b*; *Mehta et al., 2018*). While a diverse set of potential players have been identified, a clear understanding of the mechanisms that render preselection DP thymocytes sensitive to low-affinity self-ligands remains elusive.

To understand how self-reactivity shapes TCR tuning during thymic development, we compared the positive selection of thymocytes with MHC-I-restricted TCRs of low, intermediate, or high self-reactivity. We show that positive selecting signals in thymocytes with low self-reactivity occur with a more transient calcium flux and without a pronounced migratory pause. In addition, thymocytes with low self-reactivity proceed through positive selection more slowly, with a substantial fraction taking over a week to complete positive selection. Furthermore, we show that cells with lower self-reactivity retain higher expression of a 'preselection' gene expression program as they mature. This gene set including genes known to modulate TCR signals and a novel set of ion channels genes. These results

indicate that T cell tuning during thymic development occurs via developmentally layered sets of T cell tuning genes, maximizing the TCR signaling potential of thymocytes with low self-reactivity throughout their development.

## Results

### Altered TCR signaling kinetics during positive selection of thymocytes with low self-reactivity

To investigate how self-reactivity impacts CD8 SP development, we used TCR transgenic (TCRtg) mice expressing MHC-I-restricted TCRs of low, intermediate, or high self-reactivity. Mice expressing rearranged TCR α and β transgenes from a T cell clone (TG6) with low self-reactivity exhibit a strong skewing to the CD8 lineage even in a *Rag2*-sufficient background (*Figure 1—figure supplement 1*). This is indicative of robust positive selection and is in contrast with the HY TCR transgenic model where low self-reactivity is accompanied by extensive endogenous receptor rearrangement and weak skewing to the CD8 lineage (*Azzam et al., 2001*; *Kisielow et al., 1988*). Compared to CD8SP from wild-type mice, CD8SP thymocytes from TG6 mice express low levels of CD5, a reliable surface marker of T cell self-reactivity (*Figure 1a,b*; *Azzam et al., 1998*; *Mandl et al., 2013*; *Fulton et al., 2015*; *Persaud et al., 2014*; *Azzam et al., 2001*). CD8SP thymocytes from TG6 mice also have relatively low expression of Nur77, a marker of recent TCR signaling that also correlates with self-reactivity (*Figure 1c*; *Fulton et al., 2015*). In contrast, CD8SP from two well-characterized TCRtg models, OT-1 and F5, have high and intermediate levels of these markers, respectively (*Figure 1a–c*; *Fulton et al., 2015*; *Cho et al., 2016*; *Dong et al., 2017*; *Kieper et al., 2004*; *Palmer et al., 2011*; *Ge et al., 2004*). The relative levels of CD5 and Nur77 in TCRtg lymph node T cells follow a similar pattern, except that expression of Nur77 in F5 T cells decreases somewhat in lymph node relative to thymus (*Figure 1d*). This may reflect weaker recognition of peripheral versus thymic self-

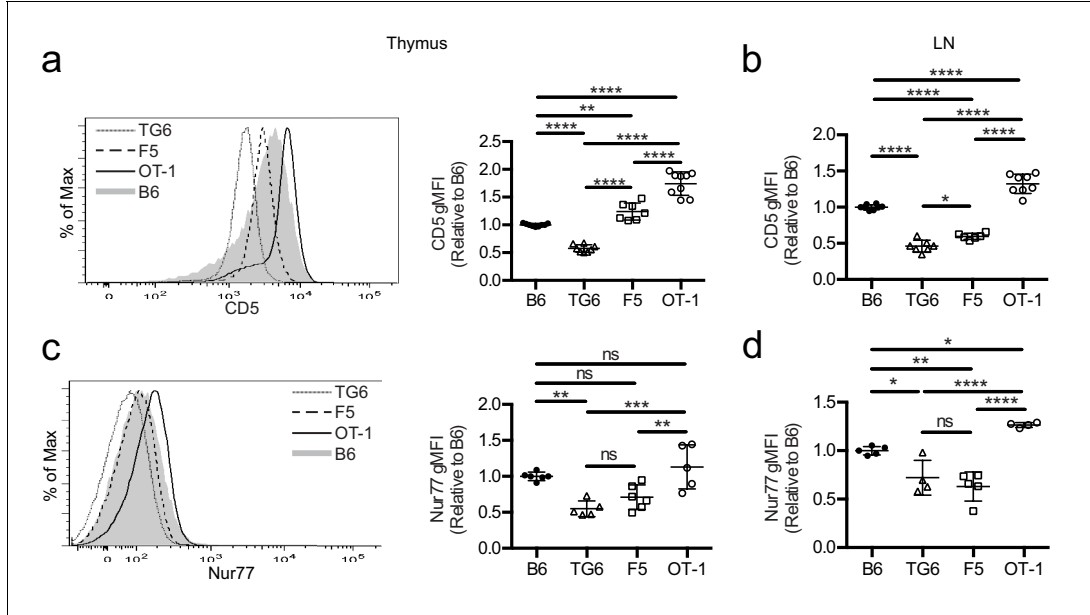

**Figure 1.** TG6, F5, and OT-1 CD8SP cells express CD5 and Nur77 across the polyclonal spectrum. CD8SP cells were harvested from the thymus and lymph nodes of wild-type (B6), TG6, F5, and OT-1 TCRtg mice and analyzed by flow cytometry. (a, b) Representative (left) and quantified by geometric mean fluorescence intensity (right) CD5 surface expression on (a) CD8SP thymocytes and (b) CD8SP lymph node cells. (c, d) Representative (left) and quantified (right) intracellular Nurr77 expression gated on (c) CD8SP thymocytes and (d) CD8SP lymph node cells. Data are presented as average ± SD and analyzed using an ordinary one-way ANOVA followed by a Tukey's multiple comparisons (*p<0.05, **p<0.01, ***p<0.001, ****p<0.0001). All data are compiled from three or more experiments.

The online version of this article includes the following figure supplement(s) for figure 1:

**Figure supplement 1.** TG6 mice have efficient positive selection and allelic exclusion in the thymus.

peptides by F5, and the loss of β5t-derived peptides in the periphery, in line with the β5t-dependence of F5 thymocytes *Nitta et al., 2010*. Thus, these three TCRtg models span the range of self-reactivity for CD8 T cell-positive selection, with TG6 providing an example of low self-reactivity coupled with relatively efficient positive selection.

Previous time-lapse studies of OT-1 thymocytes undergoing positive selection in situ revealed serial TCR signals lasting about 4 min and associated with a migratory pause (*Ross et al., 2014*; *Melichar et al., 2013*). To investigate how thymocyte self-reactivity impacts TCR signaling kinetics during positive selection, we compared the pattern of calcium changes and speed for TG6 or OT-1 thymocytes at 3 and 6 hr after the initiation of positive selection. We isolated thymocytes from TCRtg mice on a nonselecting background (herein referred to as preselection thymocytes), labeled the cells with the ratiometric calcium indicator dye Indo-1LR, overlaid the labeled cells onto selecting or nonselecting thymic slices, and imaged using two-photon, time-lapse microscopy as previously described (*Figure 2a,b*; *Bhakta and Lewis, 2005*; *Ross et al., 2014*; *Melichar et al., 2013*; *Dzhagalov et al., 2012*; *Ross et al., 2015*). For both TG6 and OT-1, the frequency of time points with elevated calcium was greater on selecting, compared to nonselecting thymic slices (*Figure 2—figure supplement 1*). This confirmed that calcium flux under these conditions is dependent on the presence of positive selecting ligands.

In line with their lower self-reactivity, TG6 cells in positively selecting slices spent slightly less time signaling than did OT-1 cells, although the difference did not reach statistical significance (*Figure 2b*, *Figure 2—figure supplement 1* and *Supplementary file 1*). To further explore this difference, we identified individual signaling events from multiple imaging runs, aligned the events based on the first time point with elevated calcium, and averaged the calcium and speed over time as previously described (*Melichar et al., 2013*) (see Materials and methods). At both 3 and 6 hr after addition to the slice, the average signaling event for TG6 thymocytes is about one minute shorter than OT-1 (*Figure 2c,d*; *Videos 1* and *2*). Moreover, while OT-1 thymocytes pause during signaling events, as reflected in a decrease in the average speed of signaling portion of the track compared to nonsignaling portion of the same cell track, TG6 thymocytes exhibit a less pronounced signal-associated pause at 3 hr, and no significant pause at the 6 hr time point (*Figure 2e*, *Figure 2—figure supplement 2*). To facilitate comparison of pausing between TG6 and OT-1 thymocytes, we calculated a 'pause index' for each signaling cell by subtracting the average speed of the signaling portion of the track from the average speed of the nonsignaling portion of the same cell track (*Figure 2f*). The pause index was significantly higher for OT-1 than for TG6 at both 3 and 6 hr (*Figure 2f*). In addition, the frequency of signaling events for TG6 thymocytes was approximately half that for OT-1 thymocytes (one signaling event every 2 hr for TG6 thymocytes versus one signaling event every 75 min for OT-1: *Supplementary file 2*). Thus, low self-reactivity is associated with less pronounced TCR-induced pausing and a reduction in both the duration and frequency of TCR signals during positive selection.

## Self-reactivity correlates with time to complete positive selection

Briefer TCR signals observed in TG6 thymocytes could mean that thymocytes with low self-reactivity need more time to accumulate sufficient signals to complete positive selection (*Au-Yeung et al., 2014*). To address this possibility, we examined the timing of different stages of positive selection using thymic tissue slice cultures (*Ross et al., 2015*). We previously showed that MHC-I-specific preselection DP thymocytes undergo a synchronous wave of maturation after introduction into positive selecting thymic tissue slices. This includes transient upregulation of the TCR activation marker CD69, followed by a switch in chemokine receptor expression from cortical (CXCR4) to medullary (CCR7) in DP thymocytes, and eventually CD4 downregulation (*Ross et al., 2014*; *Melichar et al., 2013*). To examine the impact of self-reactivity on the timing of these landmarks in positive selection, we overlaid preselection TG6, F5, or OT-1 thymocytes onto selecting and nonselecting thymic slices and analyzed progress through positive selection by flow cytometry after 2, 24, 48, 72, and 96 hr of thymic slice culture. Slice-developed TG6, F5, or OT-1 CD8SPs exhibit the same relative CD5 levels as their in vivo counterparts (*Figure 3a*), indicating that thymic slices culture provide a good model for examining the impact of self-reactivity on positive selection. In all three models, thymocytes exhibited a transient increase in CD69 expression; however, cells with lower self-reactivity reached peak CD69 expression later than cells with higher self-reactivity (*Figure 3b*). In addition, the switch in cortical to medullary chemokine receptor expression (CXCR4−CCR7+) and the appearance

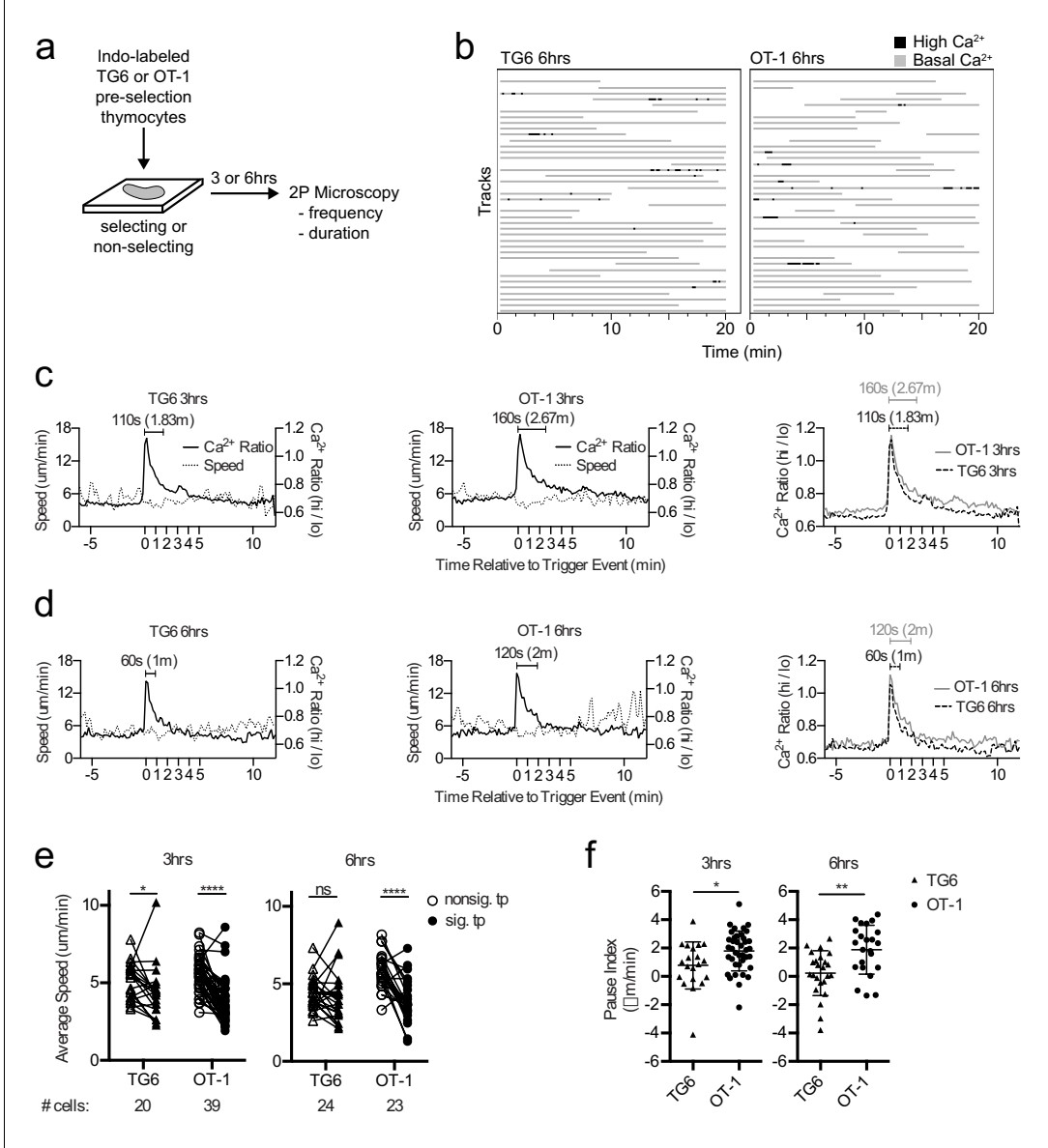

**Figure 2.** TG6 thymocytes experience briefer TCR signals and a less pronounced signal-associated migratory pause compared to OT-1 during positive selection. (a) Experimental schematic. Nonselecting slices were $\beta 2M^{-/-}$ for OT-1 experiments and either B6 ($H2^{b/b}$) or $\beta 2M^{-/-}$ for TG6 experiments. (b) Individual tracks (one track = 1 cell; each track is a horizontal line) over time from TG6 (left) and OT-1 (right) representative runs (movies) 6 hr post-addition to the slice. Gray indicates low $Ca^{2+}$ time points; black indicates high $Ca^{2+}$ time points. (c, d) For each of the indicated conditions, multiple signaling cell tracks were aligned by the beginning of the signaling event (defined by the initial rise in $Ca^{2+}$, time = 0), and the average calcium ($Ca^{2+}$) ratio and speed at each time point relative to the beginning of the signaling event are displayed. Left and center panels show overlays of $Ca^{2+}$ ratio and speed for each TCRtg, and right plots show overlay of $Ca^{2+}$ ratio for both TCRtg for comparison. (c) Average $Ca^{2+}$ ratio and speed 3 hr post-addition to the slice (TG6 n = 27 cells, OT-1 n = 37 cells). (d) Average $Ca^{2+}$ ratio and speed 6 hr post-addition to the slice (TG6 n = 25 cells, OT-1 n = 23 cells). (e) Average speed of signaling (sig) portion and nonsignaling (nonsig) portion of all tracks containing at least one signaling event. An open symbol represents the nonsignaling portion, and the closed symbols represent the signaling portion of a single track and the lines connect data from the same track. (f) The pause index for each cell in (c) was calculated by subtracting the average speed of the signaling portion of a track from the average speed of the nonsignaling portion of the same cell track. All data are compiled from two or more experiments, except OT-1 3 hr data, which is from one experiment. Preselection thymocyte populations were obtained from TG6 $Rag2^{-/-}$ $H2^b$ mice and irradiated $\beta 2M^{-/-}$ mice reconstituted with OT-1 $Rag2^{-/-}$ bone marrow. Data are presented as average ± SD and analyzed using (e) a two-way ANOVA with Sidak's multiple comparisons or (f) a Student's T-test (*p<0.05, **p<0.01, ****p<0.0001).

The online version of this article includes the following figure supplement(s) for figure 2:

**Figure supplement 1.** Percentage of signaling tracks and signaling time points in TG6 versus OT1 thymocytes at 3 and 6 hr post-addition to thymic slices.

*Figure 2 continued on next page*

Figure 2 continued

**Figure supplement 2.** TG6 thymocytes have a less pronounced pause while TCR signaling compared to OT-1 thymocytes.

of CD8SP were delayed in thymocytes with lower self-reactivity (*Figure 3c,d*, *Figure 3—figure supplement 1* and *Figure 3—figure supplement 2*). Thus, the timing of all three major landmarks during positive selection is inversely correlated with self-reactivity, implying that cells with lower self-reactivity progress more slowly through positive selection.

Mature CD8SP thymocytes appear around the end of gestation in mice and gradually accumulate after birth. If thymocytes with low self-reactivity proceed more slowly through positive selection, they may also appear more slowly during the neonatal period. To test this prediction, we performed flow cytometry on the thymus and spleen from TG6, F5, and OT-1 mice at 7, 14, and 21 days after birth. Consistent with thymic slice data, CD8SP thymocytes accumulate slowly with age in TG6 mice, with the percent of CD8SP remaining significantly below maximal adult levels at 21 days after birth (*Figure 4a and b*). In contrast, in OT-1 mice, CD8SP thymocytes reach their maximal levels at 7 days after birth, and F5 mice show an intermediate rate of accumulation (*Figure 4a and b*). In addition, TG6 mice were slowest to accumulate CXCR4-CCR7 + CD4+CD8+DP and CD8SP thymocytes (*Figure 4c*) and exhibited delayed appearance of CD8SP cells in the spleen (*Figure 4—figure supplement 1*). These results are consistent with results from thymic slice cultures and provide in vivo confirmation that the time to complete positive selection correlates inversely with thymocyte self-reactivity.

To examine the timing of positive selection in vivo in adult mice, we tracked a cohort of thymocytes using pulse labeling with the thymidine analog 5-ethynyl-2′-deoxyuridine (EdU). In vivo EdU injection labels a cohort of thymocytes proliferating after a successful TCRβ chain rearrangement at the time of the injection, which then can be detected by flow cytometry, as they undergo positive selection in vivo (*Figure 5a*; *Lucas et al., 1993*). EdU incorporated equally into the thymocytes of all three TCRtgs (*Figure 5b*). While the percent of DP cells was greater than the percent of CD8SP cells within the EdU+ population in all three transgenics at 2 days post-injection (p.i.) (*Figure 5c*), EdU + DP cells in TG6 mice remained higher than EdU + CD8 SP cells until day 6 p.i. For F5 thymocytes, EdU + CD8 SP overtook the DPs 1 day earlier (day 5 p.i.), and EdU + CD8 SP OT-1 thymocytes accumulated even more quickly – by day 4 p.i. (*Figure 5c*). The percentage of EdU + CD8 SP thymocytes in OT-1 and F5 samples exhibits a clear plateau by day 6 and reaches half-maximal EdU + CD8 SP development at day 3.5 and 4.5, respectively. In contrast, the percentage of EdU + CD8 SP thymocytes in TG6 samples continue to rise 9 days after the EdU pulse (*Figure 5c*). This data shows that increased time to complete positive selection is also correlated with low self-reactivity in vivo. Furthermore, these results show that some CD8SP thymocytes of low self-reactivity can take longer than 7 days to complete positive selection.

To confirm the relationship between self-reactivity and time to complete positive selection, we also performed EdU pulse experiments in non-TCRtg mice. We pulsed wild-type mice with EdU and then examined CD5 levels on EdU +mature TCRβ+CD8 and CD4SP thymocytes at 4, 7, and 10 days p.i. (*Figure 5d*). We observed that EdU+mature thymocytes found at earlier time points (and that therefore completed positive selection earlier) have higher CD5 compared to EdU +mature thymocytes found at later time points (*Figure 5e,f*). This is true for TCRβ +CD8 SP thymocytes; however, the trend is even more dramatic in CD4SPs (*Figure 5f*). As

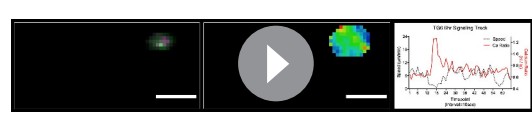

**Video 1.** Representative TG6 thymocyte signaling event. Preselection TG6 thymocytes were loaded with the ratiometric calcium indicator dye Indo1LR and allowed to migrate into selecting thymic slices for 6 hr, then imaged using two-photon-microscopy. (Left) Max z-projection of calcium high in magenta and calcium low in green. (Center) Max z-projection of the ratio of fluorescent signal in the calcium-bound channel over calcium-unbound channel for Indo1LR displayed as a heatmap (red = calcium high, purple = calcium low). (Right) Calcium ($Ca^{2+}$) ratio and speed (averaged over two intervals) of the signaling cell. Arrow indicates time points in which the cell displays elevated calcium (130 s). Frames were collected every 10 s for 10.3 min. Dimensions: 45.7 μm (width) by 75.0 μm (height) by 24.9 μm (depth). Scale bar is 10 μm.
https://elifesciences.org/articles/65435#video1

**Video 2.** Representative OT-1 thymocyte signaling event. Preselection OT-1 thymocytes were loaded with the ratiometric calcium indicator dye Indo1LR and allowed to migrate into selecting thymic slices for 6 hr, then imaged using two-photon-microscopy. (Left) Max z-projection of calcium high in magenta and calcium low in green. (Center) Max z-projection of the ratio of fluorescent signal in the calcium-bound channel over calcium-unbound channel for Indo1LR displayed as a heatmap (red = calcium high, purple = calcium low). (Right) Calcium ($Ca^{2+}$) ratio and speed (averaged over two intervals) of the signaling cell. Arrow indicates time points in which the cell displays elevated calcium (230 s). Frames were collected every 10 s for 14.3 min. Dimensions: 68.9 μm (width) by 49.8 μm (height) by 18 μm (depth). Scale bar is 10 μm.

https://elifesciences.org/articles/65435#video2

expected, we did not observe changes in CD5 expression with day post-EdU injection in TCRtg mice (*Figure 5—figure supplement 1*). This confirms that the lower CD5 expression for polyclonal thymocytes completing positive selection at later time points reflects a shift in the repertoire toward TCRs with lower self-reactivity, rather than changes in maturation state. These data indicate that self-reactivity and time to complete positive selection are also inversely correlated in nontransgenic mice, and this relationship exists for both MHC-I-restricted and MHC-II-restricted thymocytes.

## Thymocytes with low self-reactivity retain a preselection gene expression pattern marked by elevated expression of ion channel genes

Positive selection is accompanied by large-scale changes in gene expression, and delayed progress through positive selection for thymocytes with low self-reactivity might lead to synchronous or asynchronous delays in these gene expression changes. To investigate this question, we performed bulk RNA sequencing (RNA-seq) on thymocytes from TG6, F5, and OT-1 mice after sorting into three stages of development: early positive selection (CD4+CD8+CXCR4+CCR7−), late positive selection (CD4+CD8+CXCR4-CCR7+), and mature CD8SP (CD4−CD8+CXCR4-CCR7+). TG6 and OT-1 thymocytes showed significant differences in gene expression at each developmental stage, with the number of differentially expressed genes decreasing with maturity (*Figure 6—figure supplement 1*). Gene set enrichment analysis (GSEA) revealed that gene sets whose enrichment was associated with high self-reactivity (i.e., OT-1 >F5>TG6) were related to ribosome function, RNA processing, and translation in early and late positive selection thymocytes (*Figure 6a*, top, and *Figure 6—figure supplement 1*, red dots). Genes involved in cell division (*Figure 6—figure supplement 1*, purple dots) and effector function (*Figure 6—figure supplement 1*, blue dots) were also upregulated in mature CD8SP thymocytes with high self-reactivity (*Figure 6a*). Intriguingly, gene sets whose enrichment was associated with low self-reactivity (i.e., TG6 >F5>OT-1) were related to transmembrane ion channel function – particularly voltage-gated ion channels (*Figure 6b*, *Figure 6—figure supplement 1*, green dots). Genes contributing toward the enrichment scores of ion channel-related gene sets, or leading-edge genes, included components of calcium (*Cacna1e*, *Cacnb1*, *Cacnb3*, *Cacng4*), potassium (*Kcna2*, *Kcna3*, *Kcnh2*, *Kcnh3*), sodium (*Scn2b*, *Scn4b*, *Scn5a*), and chloride (*Clcn2*) ion channels.

To obtain a broader perspective on the relationship between thymocyte self-reactivity and ion channel gene expression, we curated a list of 56 ion channel genes expressed in thymocytes and examined expression changes associated with thymocyte maturation stage and self-reactivity (*Figure 6c*, *Figure 6—figure supplement 2a*). At all three developmental stages, there were more ion channel genes with higher gene expression in TG6 (green bars) compared to OT-1 (red bars) thymocytes (*Figure 6—figure supplement 2a*). Hierarchical clustering based on gene expression patterns across all samples revealed a group of genes (group 2a) that shows a strong association with low self-reactivity, and is also upregulated during early positive selection within each mouse strain. This cluster includes *Scn4b* (*Figure 6—figure supplement 2b*), which encodes the regulatory subunit of a VGSC linked to positive selection (*Lo et al., 2012*). Groups 1, 2b, and 3 are also upregulated in cells with low self-reactivity, but their expression peaks later during positive selection. Eleven of the 56 genes (group 4) have higher expression in cells with high self-reactivity and also peak at the mature CD8SP stage. This group includes *P2r × 7* and *Trmp4*, both of which have been linked to modulation of T cell effector responses (*Trebak and Kinet, 2019*; *Feske et al., 2015*).

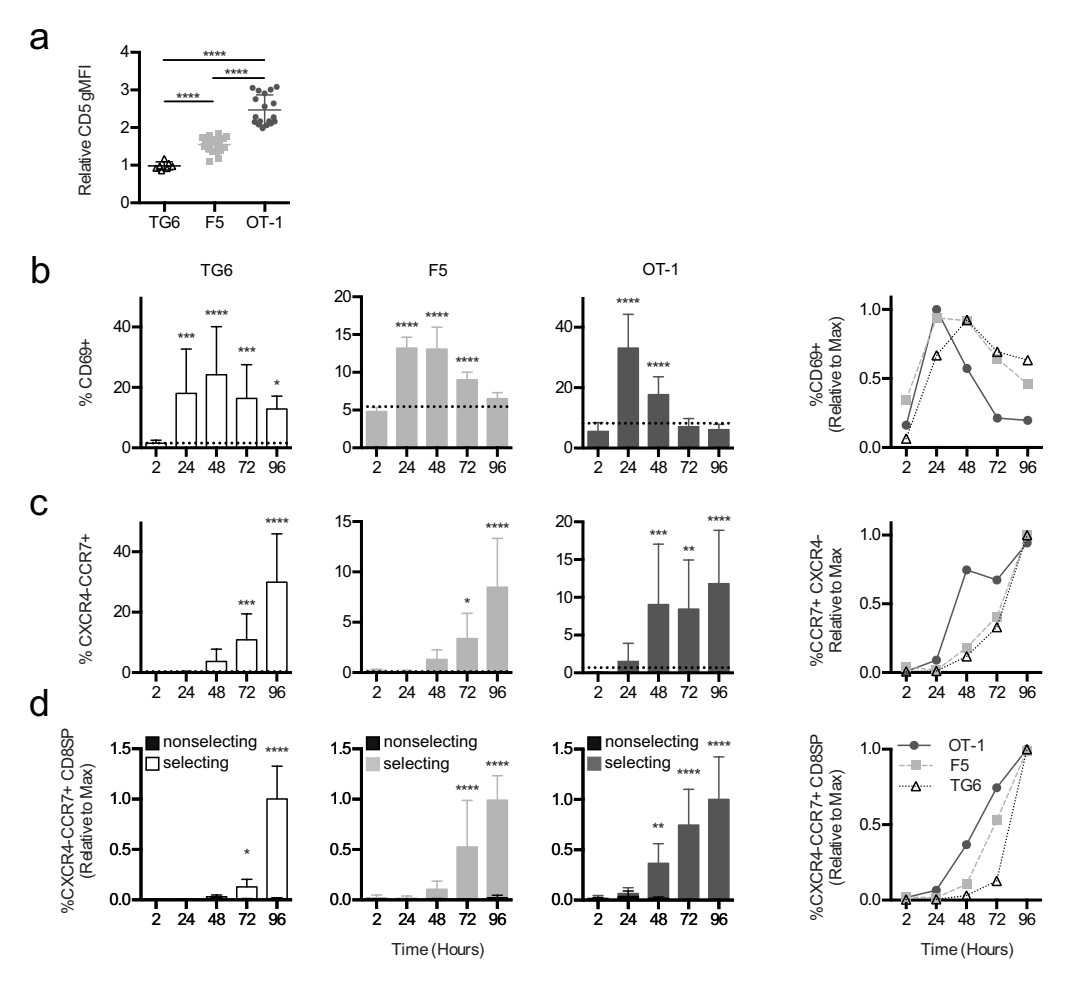

**Figure 3.** Thymocytes with lower self-reactivity exhibit delayed in situ positive selection kinetics relative to those with high self-reactivity. (a) Expression of CD5 on CXCR4−CCR7 + CD4 CD8+TG6 (96 hr), F5 (72 hr and 96 hr), and OT-1 (72 hr and 96 hr) donor cells developed in thymic slices. Values for each experiment are normalized to the average CD5 expression of the CXCR4−CCR7 + CD4 CD8+polyclonal slice resident (SR) population at the same time points. (b–d) Preselection TG6 (white bars, triangle), F5 (gray bars, square), or OT-1 (dark grey bars, circle) thymocytes were overlaid onto selecting or nonselecting thymic slices, harvested at 2, 24, 48, 72, or 96 hr post-thymocyte overlay, and analyzed using flow cytometry. Left three graphs show individual values for each transgenic. Horizontal lines in (a) and (b) indicate the average value for nonselecting slices. Right line graph shows the average for all three transgenics overlaid. (b) Percent of CD69+ cells within the CD4+CD8+ and CD4−CD8+populations. (c) Percentage of CXCR4−CCR7 + cells within CD4+CD8+ and CD4−CD8+populations. (d) Percentage of CXCR4−CCR7 + CD4 CD8+ cells out of the donor population. Values for each experiment are normalized to the average at the time point with the maximum CD8SP development (72 hr or 96 hr). For (a), data are presented as average ± SD and analyzed using an ordinary one-way ANOVA with Tukey's multiple comparisons (****p<0.000). For (b–d), data are presented as average ± SD and analyzed using an ordinary one-way ANOVA with Dunnett's multiple comparisons (**p<0.01, ***p<0.001, and ****p<0.0001) comparing to 2 hr time point. All data are compiled from three or more experiments. Preselection thymocyte populations were obtained from TG6tg Rag2$^{-/-}$ H2$^b$ mice, irradiated β2M$^{-/-}$ mice reconstituted with F5tg Rag1$^{-/-}$ bone marrow, and OT-1tg Rag2$^{-/-}$ β2M$^{-/-}$ mice. See also *Figure 3—figure supplements 1* and *2*.

The online version of this article includes the following figure supplement(s) for figure 3:

**Figure supplement 1.** TG6 thymocytes exhibit delayed development in thymic slices.

**Figure supplement 2.** TG6 thymocytes have delayed expression of positive selection markers in thymic slices.

Many ion channel genes whose expression correlated with low self-reactivity were also preferentially expressed at the early positive selection stage (*Figure 6c*, group 2a), suggesting that thymocytes of low self-reactivity may retain an immature gene expression pattern later into positive selection. To further investigate this question, we identified a gene signature associated with preselection DP thymocytes using microarray data from the Immunological Genome Project

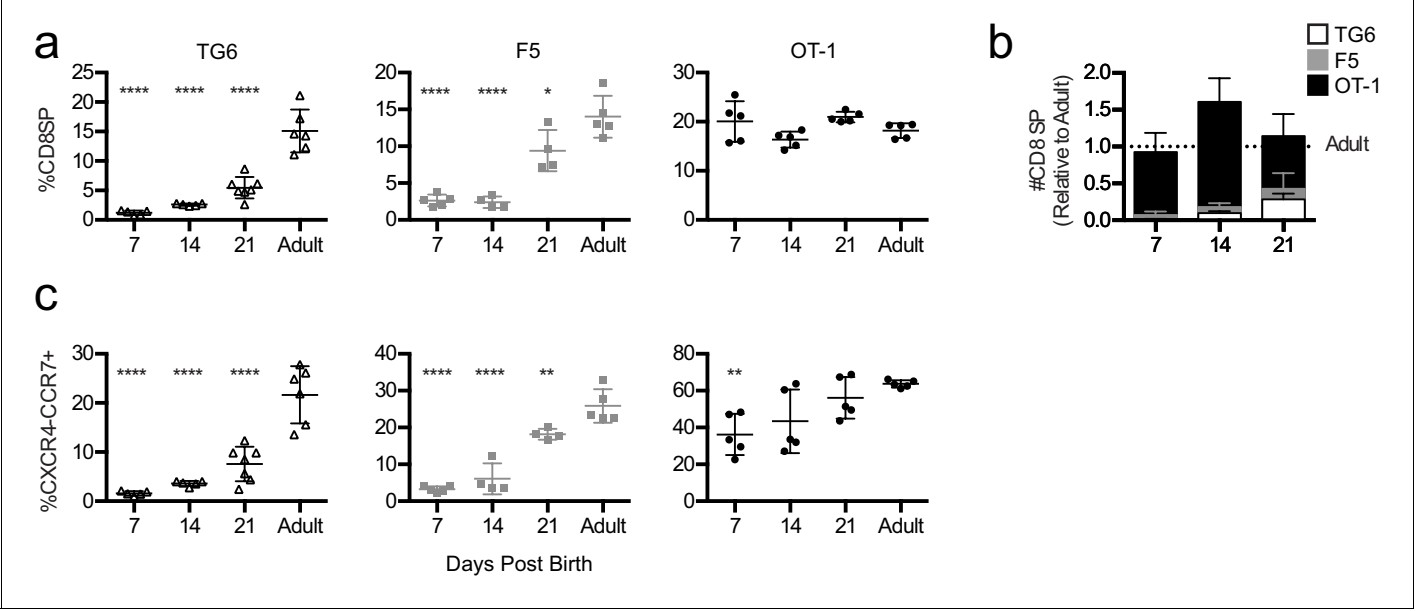

**Figure 4.** Mature thymocytes with lower self-reactivity appear later post-birth than those with higher self-reactivity. Thymuses were harvested from TG6 (triangle, white bar), F5 (square, gray bar), or OT-1 (circle, black bar) transgenic neonatal mice at 7, 14, and 21 days post-birth and from adult mice (6–9 weeks old), and analyzed using flow cytometry. (a) Percentage of CD4−CD8+ cells. (b) Total number of CD4−CD8+ thymocytes at the indicated time point post-birth, relative to adult, for each transgenic. (c) Percentage of CXCR4−CCR7+ cells within CD4+CD8+ and CD4−CD8+ populations. Data are presented as average ± SD and analyzed using an ordinary one-way ANOVA, Dunnett's multiple comparisons (*p<0.05, **p<0.01, and ****p<0.0001) comparing to adult. All data are compiled from three or more experiments.

The online version of this article includes the following figure supplement(s) for figure 4:

**Figure supplement 1.** T cells with lower self-reactivity appear later post-birth than those with higher self-reactivity in the spleen.

(ImmGen) (*Heng et al., 2008*). Specifically, we compared non-TCR transgenic, preselection (CD69−) DP thymocytes to their immediate progeny (CD69+) DP thymocytes, and to their immediate precursor (DN4 thymocytes), and identified a set of 11 genes using the dual criteria of high expression in CD69− DP, and largest fold increase in CD69−DP relative to both DN4 and CD69+DP (see Materials and methods) (*Figure 6d*, top, red dots). We then examined expression of these 11 genes in TG6, F5, and OT-1 thymocytes at different developmental stages (*Figure 6d*, bottom). Interestingly, all 11 genes correlated with low self-reactivity, particularly at the early DP stage of development. This included *Themis* (*Figure 6—figure supplement 2b*), which encodes a protein required for modulating TCR signaling during positive selection (*Lesourne et al., 2009*; *Patrick et al., 2009*; *Kakugawa et al., 2009*; *Fu et al., 2009*; *Johnson et al., 2009*; *Choi et al., 2017b*; *Choi et al., 2017a*; *Mehta et al., 2018*). Moreover, expression of the curated set of ion channel genes in the ImmGen microarray data set revealed that group 2 genes, whose expression correlated with low self-reactivity, also displayed higher expression in preselection wild-type thymocytes (*Figure 6—figure supplement 2c*). Furthermore, GSEA between CD69− DP compared to DN4 or CD69+DP thymocytes showed enrichment of gene sets relating to ion channels (DN4<CD69 DP>CD69+DP) (*Figure 6—figure supplement 2d*). Together, this data suggests that cells with low self-reactivity retain a preselection DP phenotype later into development than those with high self-reactivity and that elevated expression of ion channel genes is a prominent feature of that phenotype.

Retention of a preselection gene expression program can account for some, but not all, of the elevated ion channel gene expression by thymocytes of low self-reactivity. Specifically, there is another set of ion channel genes whose expression peaks at the late positive selection or CD8SP stage (*Figure 6c*, group 3). Interestingly, two of these genes, *Kcna2* and *Tmie* (*Figure 6—figure supplement 2b*), are also upregulated in mature peripheral CD5 low CD8SP T cells relative to CD5 high CD8SP T cells, based on published data sets (*Supplementary files 3* and *4*; *Fulton et al., 2015*; *Matson et al., 2020*). Additionally, cells with low self-reactivity also have high expression of the sodium-dependent neutral amino acid transporter *Slc6a19* (*Figure 6—figure supplements 1* and

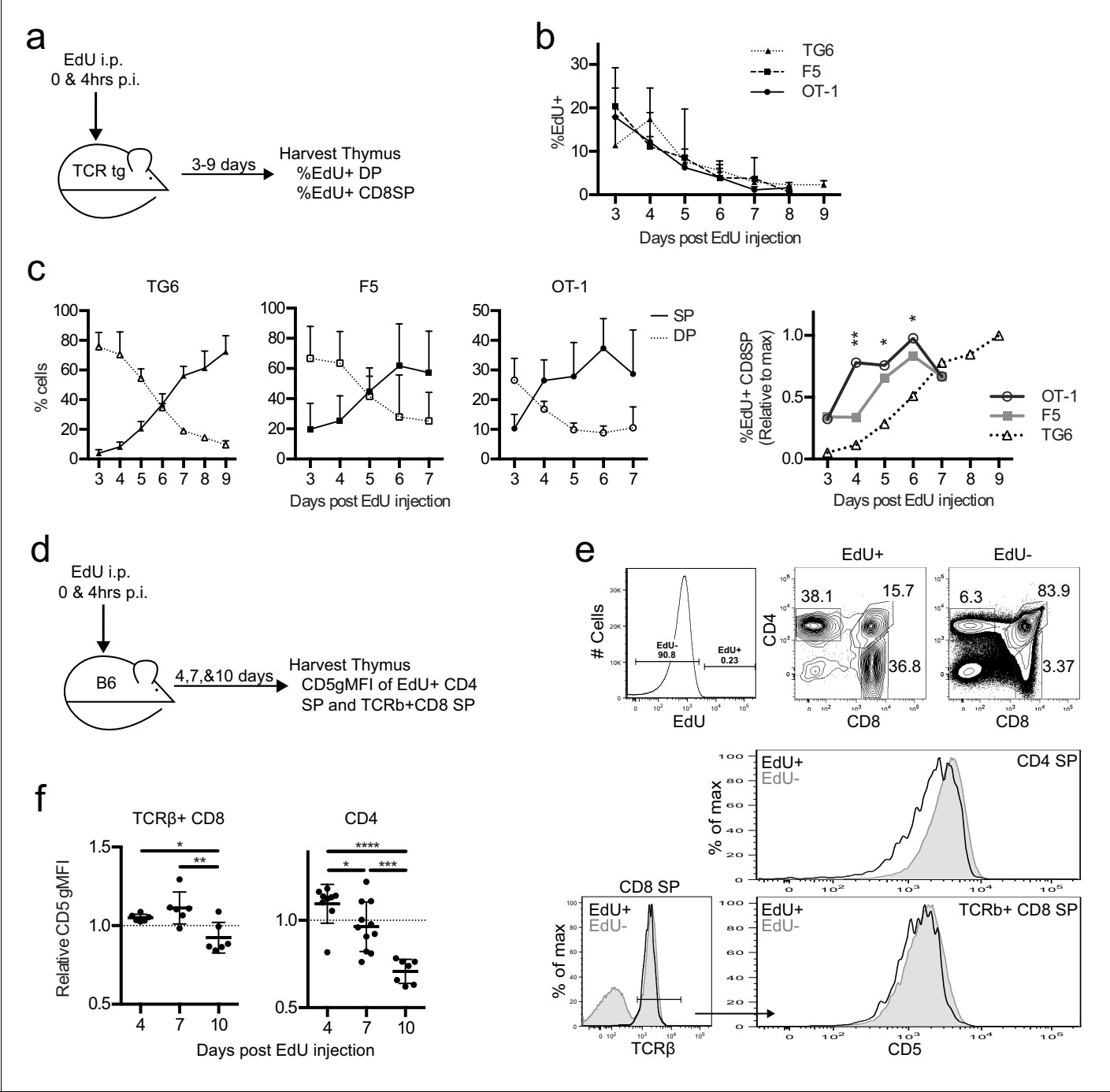

**Figure 5.** Thymocytes with lower self-reactivity have delayed development in steady state TCRtg and wild-type mice. TG6, F5, or OT-1 (**a–c**) or B6 (**d–f**) mice were injected with two doses of 1 mg of EdU intraperitoneally (i.p.) at 0 and 4 hr post-injection (p.i.). (**a**) TCRtg experimental schematic. (**b**) EdU incorporation into TCRtg thymuses. (**c**) (Left three graphs) Percent CD8SP and DP thymocytes out of gated EdU+ cells at various days p.i. in TG6, F5, and OT-1 tg thymuses. (Right) The percent of EdU+CD8 SP relative to max within each experiment. Data are presented as the average within each transgenic and analyzed using an ordinary one-way ANOVA at each time point (*p<0.05, **p<0.01). Tukey's multiple comparisons showed significant differences between OT-1 and TG6 at day 4 (**), day 5 (*), and day 6 (*); OT-1 and F5 at day 4 (*); and a difference between F5 vs TG6 at Day 6 (p=0.069). (**d**) B6 experimental schematic. (**e**) Representative gating strategy and histograms showing B6 day 10 p.i. CD5 expression on EdU+ and EdU− mature SP cells. (**f**) CD5 expression of EdU+TCRβ+CD8 SP and EdU+CD4 SP thymocytes at 4, 7, and 10 days p.i. CD5 expression shown relative to total TCRb+CD8 SP or CD4SP CD5 expression. In (**f**), data are presented as average ± SD and analyzed using an ordinary one-way ANOVA, Tukey's multiple comparisons (*p<0.05, **p<0.01, ***p<0.001, ****p<0.0001). All data are compiled from three or more experiments.

The online version of this article includes the following figure supplement(s) for figure 5:

*Figure 5 continued on next page*

*Figure 5 continued*

**Figure supplement 1.** CD5 expression does not change in TCRtg EdU+CD8 SP, regardless of day post-EdU injection.

2b, *Supplementary files 3* and *4*; *Fulton et al., 2015*; *Matson et al., 2020*). Thus, elevated expression of distinct sets of ion channel genes is observed throughout the development of CD8 T cells with low self-reactivity.

## Discussion

T-cell-positive selection is a multi-day process during which thymocytes experience transient, serial TCR signals, triggering a series of phenotypic changes culminating in co-receptor downregulation and lineage commitment. Preselection DP thymocytes are highly sensitive to low-affinity self-ligands, and gradually downmodulate their sensitivity as they mature (*Davey et al., 1998*; *Lucas et al., 1999*). However the factors that modulate TCR sensitivity during positive selection are not well understood. Moreover, there is a growing appreciation that thymocytes differentially adjust their response to self-ligands based on their degree of self-reactivity, although little is known about how positive selection differs for thymocytes of relatively low versus relatively high self-reactivity. Here we show that thymocytes with low self-reactivity experience briefer TCR signals and can take more than twice as long to complete positive selection, compared to those with relatively high self-reactivity. Thymocytes with low self-reactivity retain a preselection gene expression program as they progress through development, including elevated expression of genes previously implicated in modulating TCR signaling during positive selection, and a novel set of ion channel genes. In addition, a separate set of ion channel genes that peaks at the CD8SP stage is also selectively upregulated in mature CD8SP thymocytes of low self-reactivity. These results indicate that the modulation of TCR sensitivity during T cell development occurs with distinct kinetics for T cells with low self-reactivity and associates membrane ion channel expression with early and late stages of T cell development as well as T cell self-reactivity.

Our results contrast with an earlier study, which reported that the overall time for positive selection was constant for different CD8 T cell clones, with an accelerated early phase compensating for a delayed late phase for CD8 T cells of low self-reactivity (*Kimura et al., 2016*). Importantly, that study relied on the decrease in Rag-GFP reporter expression to infer the timing of development, rather than directly tracking cohorts of positively selecting cells as we have done in the current study. While the slow loss of Rag-GFP reporter due to GFP protein turnover can provide an indication of the elapsed time after shutoff of rag expression (*McCaughtry et al., 2007*), reporter expression during the early phase of positive selection may be impacted by the timing of rag shutoff and cell turnover, making this an unreliable indicator of developmental time in this experimental setting.

A variety of molecules have been implicated in modulating the TCR sensitivity of DP thymocytes, including the TCR-associating protein Themis, ER-associating protein Tespa1, microRNA mir-181a, and components of a VGSC (encoded by *Scn4b* and *Scn5a*) (*Wang et al., 2012*; *Lo et al., 2012*; *Fu et al., 2009*; *Johnson et al., 2009*; *Lesourne et al., 2009*; *Patrick et al., 2009*; *Kakugawa et al., 2009*). Here we show that the VGSC is part of a larger group of ion channel genes, that also includes components of voltage-gated calcium channels (*Cacna1e, Cacnb3*), that are both upregulated in preselection DP thymocytes, and selectively retained during the positive selection of thymocytes with low self-reactivity. We also identified another group of ion channels whose expression peaks at the CD8SP stage, and that are also upregulated in CD8 T cells of low self-reactivity. This group includes *Kcna2,* a component of a voltage-gated potassium channel, and *Tmie*, a gene essential for calcium-dependent mechanotransduction in hair cells within the inner ear (*Qiu and Müller, 2018*; *Zhao et al., 2014*). Thus, ion channel expression appears to be selectively modulated based on the degree of self-reactivity at two distinct stages of T cell development and may help to enhance weak TCR signals both during the initial phase of positive selection and during T cell homeostasis in the periphery.

Voltage-gated ion channels are best known for their role in generating action potentials in neurons and muscle cells, but they also play roles in nonelectrically excitable cells such as T cells and thymocytes (*Feske et al., 2015*; *Lo et al., 2012*; *Cahalan and Chandy, 2009*; *Trebak and Kinet,*

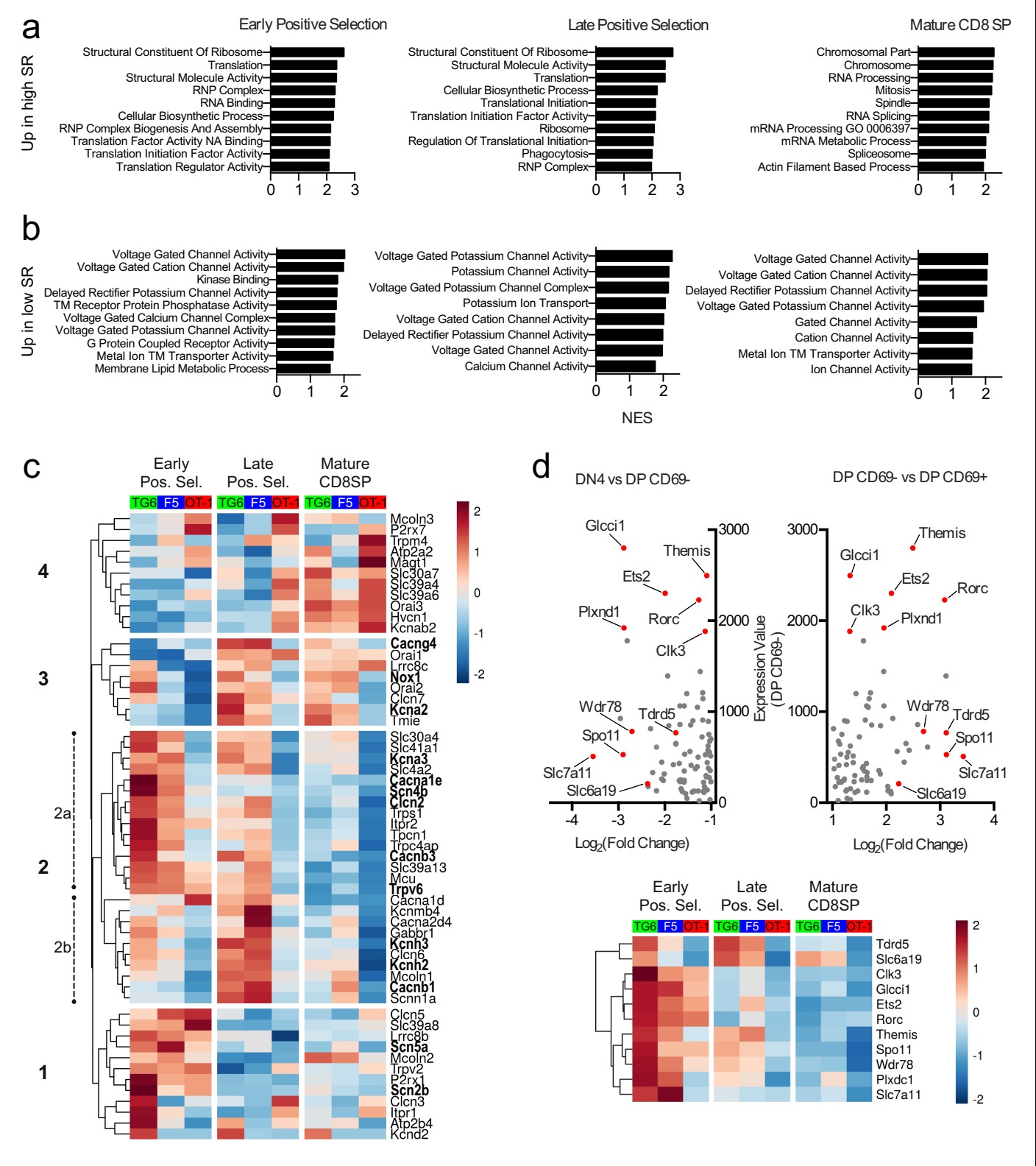

**Figure 6.** Thymocytes with low self-reactivity retain a preselection gene expression program marked by elevated expression of ion channel genes. RNA-seq of OT-1, F5, and TG6 thymocytes at the early positive selection, late positive selection, and mature CD8SP stages. All samples have three biological replicates except for TG6 late positive selection, which has two biological replicates. (a, b) Gene set enrichment analysis (GSEA) was performed on all pairwise combinations of TCRtg samples, and the gene sets were filtered for enrichment in the order of self-reactivity (OT-1>F5>TG6)

*Figure 6 continued on next page*

*Figure 6 continued*

(a) or in the inverse order of self-reactivity (TG6>F5>OT-1) (b). Normalized enrichment scores (NES) are from the OT-1 vs TG6 (a) or TG6 vs OT-1 (b) comparison. Abbreviations: self-reactive (SR), ribonucleoprotein (RNP), nucleic acid (NA), transmembrane (TM). (c) Heatmap of differentially expressed (padj<0.05 for OT-1 vs TG6 comparison) ion channel genes (see Materials and methods) across all three stages and all three transgenic models. Gene expression values normalized by the DESeq2 variance stabilizing transformation were averaged over biological replicates and scaled per row. Heatmap is hierarchically clustered into four groups, numbered in the order of expression during development. Leading-edge genes from GSEA are in bold. (d) (Top) Plot of expression level in DP CD69− thymocytes versus fold difference for DN4 versus DP CD69− (left side of X-axis) or DP CD69− versus DP CD69+ (right side of X-axis) for ImmGen microarray data. The 11 genes chosen to represent the 'preselection DP' gene signature are indicated with red dots. (Bottom) Heatmap of normalized expression (as in (c)) of the 11 representative preselection DP genes in TG6, F5, and OT-1 thymocytes at the indicated stage of development.

The online version of this article includes the following figure supplement(s) for figure 6:

**Figure supplement 1.** Gene expression differences between TG6 and OT-1 thymocytes at different stages of development.

**Figure supplement 2.** Thymocytes with low self-reactivity retain a preselection gene expression program marked by elevated expression of ion channel genes.

---

*2019*). In these cells, voltage-gated ion channels have been proposed to be regulated by local fluctuations of membrane charge, or alternative mechanisms such as phosphorylation (*Rook et al., 2012*). While the link between ion channels and T cell tuning is not known, the modulation of calcium entry is a potential mechanism. TCR triggering initially leads to a rise in cytosolic calcium via release of ER stores, and the calcium rise is sustained in mature T cells when depletion of ER calcium stores triggers the opening of CRACs. However, CRACs appear to be dispensable for positive selection (*Oh-hora, 2009*; *Gwack et al., 2008*; *Vig and Kinet, 2009*), and it remains unclear how thymocytes sustain calcium signals during the days required to complete positive selection. Evidence suggests that the VGSC promotes prolonged calcium flux in response to TCR triggering in DP thymocytes, although the mechanism remains unclear (*Lo et al., 2012*). Our data indicates that, not only sodium, but also potassium, chloride, and calcium channels are upregulated in preselection DP thymocytes and positive selecting thymocytes with low self-reactivity, implying that positive selection leads to broad changes in transmembrane ion flux. In addition to potentially promoting calcium influx, it is tempting to speculate that changes in membrane ion flux could directly impact TCR triggering by regulating local membrane charge at the receptor complex (*Ma et al., 2017*). Altered ion flux in T cells with low self-reactivity may also impact ion-coupled transport of nutrients. In this regard, it is intriguing that the sodium-dependent neutral amino acid transporter gene *Slc6a19* is also consistently upregulated in thymocytes and CD8 T cells with low self-reactivity (*Figure 6c,d*, *Figure 6—figure supplement 1*, *Figure 6—figure supplement 2b* and *Supplementary files 3* and *4*; *Fulton et al., 2015*; *Matson et al., 2020*).

Taken together, our results support a model in which gene expression is altered in two, temporally distinct phases based on the degree of self-reactivity (*Figure 7*). Thymocytes with low self-reactivity (dashed lines) experience briefer TCR signals and accumulate TCR signals more slowly, resulting in a delayed time course of positive selection. During phase 1, a 'preselection DP' gene expression program (including *Themis*, *Scn4b*, and *Cacna1e*) is downregulated in all thymocytes undergoing positive selection, but more gradually and incompletely in thymocytes with low self-reactivity. During phase 2, a distinct set of genes (including *Kcna2* and *Tmie*) whose expression peaks in CD8 SP are upregulated to a greater extent in thymocytes with low self-reactivity. In combination with previously published findings, our model suggests that phase 1 and phase 2 genes help to sustain TCR signals throughout positive selection. Thus, high expression of phase 1 and phase 2 genes, early and late in development (respectively), may allow thymocytes with low self-reactivity to continue to receive survival and differentiation signals in spite of their low reactivity.

Previous work has shown that CD8 T cells can take three or more days to complete development, but it was unclear why positive selection takes longer for some cells than others (*Saini et al., 2010*; *Lucas et al., 1993*; *Kurd and Robey, 2016*). Our data show that positive selection can take considerably longer for cells with low self-reactivity compared to those with high self-reactivity. In particular, EdU pulse labeling in adult TG6 transgenic mice shows that some thymocytes are still converting from DP to CD8SP after more than 1 week of positive selection. It has been previously noted that CD8 T cells that arise in the neonatal period have higher CD5 levels and are more proliferative and responsive to cytokines compared to CD8 T cells that arise in adult mice (*Dong et al., 2017*;

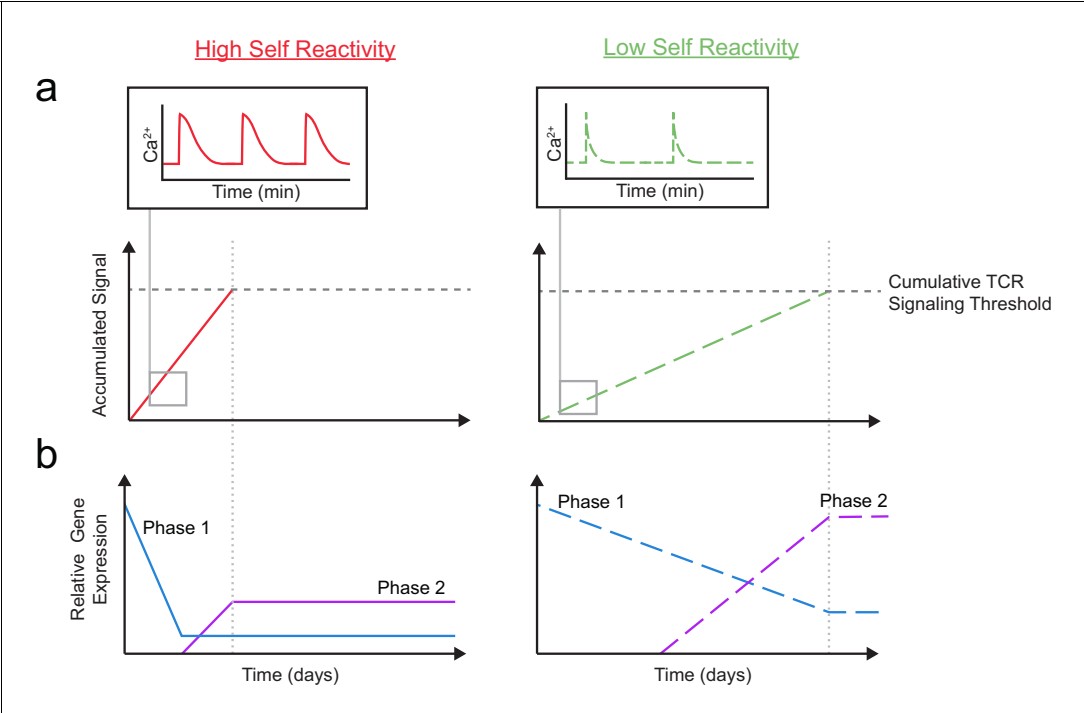

**Figure 7.** A model for temporal signal accumulation and gene expression changes during development in cells with relatively high and low self-reactivity. (a) (Left) Positively selected cells with relatively high self-reactivity (SR) (solid, red lines) receive serial, brief (minutes) TCR signals, allowing for signal accumulation over a period of days, to eventually complete positive selection. (Right) Those with relatively low self-reactivity (dashed, green lines) experience even briefer TCR signals leading to slower signal accumulation and a longer time to achieve the cumulative TCR signaling threshold required for positive selection. (b) High (left) and low (right) self-reactive thymocytes have distinct temporal regulation of early (phase 1: blue lines) and a late (phase 2: purple lines) gene expression during positive selection. Elevated expression of phase one genes (including *Themis*, *Scn4b*, and *Cacna1e*) in preselection DP thymocytes may contribute to their high TCR sensitivity for self-ligands. We propose that phase 1 genes are downregulated more slowly in thymocytes with low self-reactivity, allowing them to retain high TCR sensitivity later into development, and aiding in signal accumulation from weak ligands. We also suggest that, later in positive selection, phase 2 genes (including *Kcna2* and *Tmie*) are upregulated to a greater extent in thymocytes with low self-reactivity and could help to increase sensitivity to self-ligands late in development and in the periphery.

*Smith et al., 2018*). This phenomenon may be explained in part by the shorter time required for T cells with high self-reactivity to complete positive selection. However, distinct mechanisms independent of TCR specificity also contribute to this phenomenon. For example, T cells that arise in the neonatal period are derived from fetal stem cells and are intrinsically programmed to be more proliferative than those derived from adult stem cells (*Smith et al., 2018*; *Wang et al., 2016*; *Mold et al., 2010*; *Ikuta et al., 1990*; *Bronevetsky et al., 2016*). There are also indications that age-related changes in the thymic environment may alter the TCR affinity thresholds for positive and negative selection (*Dong et al., 2017*). Thus, three distinct mechanisms may all contribute to a bias toward high self-reactivity T cells in young individuals, with T cells of lower self-reactivity accumulating with age. It is tempting to speculate that this arrangement is evolutionarily advantageous, since highly self-reactive T cells can provide a rapid response against acute infections, which are the biggest threat to a young individual. As the organism ages, however, it may become advantageous to have a more diverse, specialized T cell repertoire, particularly to fight chronic infections. In line with this notion, we recently reported that TG6 T cells are resistant to exhaustion and provide a protective response to *Toxoplasma gondii* chronic infection, properties that are linked to limited binding of self-peptides to the restricting MHC-I molecule L[d] (*Blanchard et al., 2008*; *Chu et al., 2016*; *Tsitsiklis et al., 2020*).

Overall, the work presented here provides insight on the differences in the development of T cells with low and high self-reactivity that may impact their distinct peripheral functions. The question of how a lengthy positive selection process and differences in ion channel expression help to functionally tune T cells with low self-reactivity remains an important area for future investigations.

## Materials and methods

### Mice and bone marrow chimeras

All mice were bred and maintained under pathogen-free conditions in an American Association of Laboratory Animal Care-approved facility at the University of California, Berkeley. The University of California, Berkeley Animal Use and Care Committee approved all procedures. B6 (C57BL/6, Stock No. 000664), B6.C (B6.C-H2$^d$/bByJ, Stock No. 000359), and $Rag2^{-/-}$ (B6(Cg)-$Rag2^{tm1.1Cgn}$/J, Stock No. 008449), $Rag1^{-/-}$ (B6.129S7-$Rag1^{tm1Mom}$, Stock No. 002216), and $B2M^{-/-}$ (B6.129P2-$B2M^{tm1Unc}$/DcrJ, Stock No. 002087) mice were from Jackson Labs. OT-1 $Rag2^{-/-}$ (B6.129S6-$Rag2^{tm1Fwa}$ Tg(TcraTcrb)1100Mjb, Model No. 2334) mice were from Taconic. F5 $Rag1^{-/-}$ and B6xB6C ($H2^{b/d}$) mice were generated by crossing as previously described (*Au-Yeung et al., 2014*; *Chu et al., 2016*). TG6 transgenic mice were generated as previously described by *Chu et al., 2016* and further crossed in house to generate TG6 $H2^{b/d}$ (used in neonatal, EdU pulse experiments, and RNA-seq), and TG6 $H2^{b/d}Rag2^{-/-}$ (used to measure self-reactivity), and TG6 $H2^bRag2^{-/-}$ mice (used for overlay onto thymic slices). Preselection OT-1 $Rag2^{-/-}$ thymocytes were generated by crossing onto a nonselecting MHC-I deficient background ($B2M^{-/-}$) or by transferring 1 × 10$^6$ OT-1 $Rag2^{-/-}$ bone marrow cells intravenously (i.v.) into lethally irradiated (1200 rad) $B2M^{-/-}$ recipients (used for imaging experiments). Preselection F5 $Rag1^{-/-}$ thymocytes were generated by transferring 1 × 10$^6$ bone marrow cells i.v. into lethally irradiated (1200 rad) $B2M^{-/-}$ recipients. Preselection TG6 $Rag2^{-/-}$ thymocytes were generated by crossing onto a nonselecting background lacking H2$^d$ MHC-I (B6, H2$^b$ haplotype). Bone marrow chimeras were analyzed 5–7 weeks following reconstitution. For all experiments, mice were used from 3 to 10 weeks of age, with the exception of neonatal experiments where mice were used one to nine weeks of age.

### Experimental design

Sample size in this study was consistent with previous studies and was not determined based on a prior statistical test. All experiments were completed with at least three biological replicates unless otherwise stated in the figure legend. Consistent with previous studies, three to eight thymic slices (technical replicates) per condition were used for thymic slice experiments analyzed by flow cytometry. Samples were excluded if their thymic phenotype indicated signs of stress (<10$^7$ total cells or less than 20% DP thymocytes out of live thymocytes). Samples for thymic slice experiments were excluded if the culture had very low viability (<15%) or <0.2% of donor derived thymocytes of total live cells in the final sample. For EdU experiments, samples that did not incorporate detectable EdU (<0.001% of live cells that were EdU+) were excluded. For experiments where thymic slices from the same genotypic background received multiple treatments, slices were randomly allocated to treatment conditions.

### Thymic slices

Preparation of thymic slices has been previously described (*Dzhagalov et al., 2012*; *Ross et al., 2015*). Briefly, thymic lobes were gently isolated and removed of connective tissue, embedded in 4% agarose with a low melting point (GTG-NuSieve Agarose, Lonza), and sectioned into slices of 400–500 μm using a vibratome (VT1000S, Leica). Slices were overlaid onto 0.4 μm transwell inserts (Corning, Cat. No. 353090) set in six well tissue culture plates with 1 ml cRPMI under the insert. 2 × 10$^6$ thymocytes in 10 μl cRPMI were overlaid onto each slice and allowed to migrate into the slice for 2–3 hr. Afterwards, excess thymocytes were removed by gentle washing with PBS and cultured at 37˚C 5%CO$_2$ until harvested for flow cytometry or two-photon analysis. For flow cytometry, thymic slices were dissociated to single-cell suspensions and then stained with fluorophore-conjugated antibodies. For two-photon imaging, thymic slices were glued onto a glass coverslip and imaged as described below.

### Thymocyte labeling for overlay onto thymic slices

Thymuses collected from TCRtg preselection mice and dissociated through a 70 μm cell strainer to yield a cell suspension. For overlay onto thymic slices followed by flow cytometry, TCRtg preselection thymocytes were labeled with 1 μM Cell Proliferation Dye eFluor450 or 0.5 μM Cell Proliferation Dye eFluor670 as previously described (*Kurd et al., 2019*). For two-photon imaging of thymic slices,

thymocytes were labeled with 2 mM leakage-resistant Indo-1 (ThermoFisher Scientific, Cat. No. I1226) as previously described followed by overlay onto slices (*Melichar et al., 2013*). For selecting conditions, B6 slices were used for OT-1 and F5, and B6xB6C (H2$^{b/d}$) slices were used for TG6. For nonselecting conditions, β2M$^{-/-}$ slices were used for OT-1 and F5, and either B6 (H2$^{b/b}$) or β2M$^{-/-}$ slices were used for TG6. We did not note any differences when B6 (H2$^{b/b}$) or β2M$^{-/-}$ slices were used as nonselecting slices for TG6, and data was combined.

## Flow cytometry

Thymic slices, whole thymuses, and spleens were dissociated into FACS buffer (0.5% BSA in PBS) or RPMI and passed through a 70 μm filter before staining. Splenocytes were then red blood cell lysed using ACK lysis buffer (0.15M NH$_4$Cl, 1 mM KHCO$_3$, 0.1 mM Na$_2$EDTA) for 5 min at room temperature prior to staining. For thymic slices, cells were first stained for 60 min in a 37°C water bath with CXCR4 (L276F12 or 2B11) and CCR7 (4B12) antibodies in 2.4G2 supernatant. Cells were then surfaced stained for 15 min on ice in 2.4G2 supernatant containing the following antibodies: CD4 (clone GK1.5 or RM4-5), CD8α (53–6.7), CD5 (53–7.3), and CD69 (H1.2F3). Finally, cells were washed in PBS and stained in Ghost Dye Violet 510 (Tonbo Biosciences, Cat. No. 13–0870 T100) for 10 min on ice. The following antibodies were used for flow cytometric analysis of whole thymuses and spleens: CD4 (GK1.5 or RM4-5), CD8α (53–6.7), CD5 (53–7.3), and CD69 (H1.2F3), CD24 (M1/69), TCRβ (H57-597), CXCR4 (L276F12), CCR7 (4B12), Vα2 (B20.1), Vβ8.1/8.2 (KJ16-133.18), and Vβ2 (B20.6). Cells were then washed in PBS and stained in Ghost Dye Violet 510 as described above. For intracellular staining, cells were then fixed and permeabilized using Invitrogen's Transcription Factor Staining Buffer Set (ThermoFisher Scientific; Cat. No. 00-5523-00) according to manufacturer's instructions and stained with Nur77 (12.14) antibody (ThermoFisher Scientific, Cat. No. 12-5965-82). Cells were analyzed using an LSRII, LSR, or Fortessa X20 flow cytometer (BD Biosciences) and data analyzed using FlowJo software (Tree Star).

## EdU pulse labeling

TCRtg and nontransgenic selecting mice, aged 5–8 weeks, received two intraperitoneal (i.p.) injections of EdU (ThermoFisher Scientific, Cat. No. A10044) 1 mg each and 4 hr apart, as previously described for BrdU (*Lucas et al., 1993*). Thymuses were harvested from pulsed mice 3–10 days after the second injection and dissociated into a single-cell suspension and were surface stained with CD4, CD8α, CD5, CD69, CD24, and TCRβ antibodies as described above. Cells were then processed using Click-iT EdU Alexa Fluor 488 Flow Cytometry Assay Kit from Invitrogen according to the manufacturer's instructions (ThermoFisher Scientific, Cat. No. C10420). Flow cytometry and data analysis were performed as described above.

## Two-photon imaging

Two-photon imaging of thymic slices has been previously described (*Melichar et al., 2013*; *Dzhagalov et al., 2012*; *Ross et al., 2014*; *Dzhagalov et al., 2013*; *Au-Yeung et al., 2014*). All imaging was performed 3–6 hr after the addition of thymocytes to thymic slices. Thymic slices were glued onto glass cover slips and fixed to the bottom of a petri dish being perfused at a rate of 1 ml/min with 37°C oxygenated, phenol red–free DMEM during imaging using an MPII Mini-Peristaltic Pump (Harvard Apparatus) and a 35 mm Quick Exchange Platform (QE-1) in conjunction with a TC-324B temperature controller (Warner Instruments). Samples were imaged in the cortex of the thymus, determined by proximity to the thymic capsule. Images were collected using a Zeiss LSM 7 MP upright, two-photon microscope with a 20×/1.0 immersive Zeiss objective and a Coherent Chameleon laser tuned to 720 nm for imaging Indo-1. Ratiometric Ca$^{2+}$ signals were collected with 440-nm-long pass dichroic mirror and 400/45 and 480/50 bandpass filters. Ca$^{2+}$ hi signals (~401 nm) were collected using a bialkali PMT; Ca$^{2+}$ lo signals (~475 nm) were collected using a GaASP PMT. Image areas of up to 212 × 212 μm to a depth of up to 55 μm were acquired every 10 s for 15 or 20 min with 3 μm z steps starting from beneath the cut surface, using Zen 2010 software from Zeiss.

## Image analysis

Two-photon movies were processed and rendered using Imaris software (Bitplane Scientific Software) to collect x, y, and z coordinates and fluorescence intensity as previously

described (*Melichar et al., 2013*; *Ross et al., 2014*; *Au-Yeung et al., 2014*). Imaging data were then analyzed using a custom MATLAB script (MathWorks; as previously described [github.com/lylutes/sparky5; *lylutes, 2021*; *Ross et al., 2014*]), ImageJ, and Microsoft Excel. GraphPad Prism software was used for graphing and statistical analysis. For calculation of the percent of signaling time points, data were converted into flow cytometry-like files using custom DISCit software and further analyzed using FlowJo software (TreeStar) (*Moreau et al., 2012*). The background calcium ratio is average $Ca^{2+}$ ratio under nonselecting conditions (0.675 for OT-1 and TG6) from the $Ca^{2+}$ ratio of the raw fluorescence values at each time point per run. To calculate the average event length, cells were considered to be signaling when the average $Ca^{2+}$ ratio was $\geq$0.2 above the background calcium ratio for $\geq$2 consecutive time points on a cell track. A trigger event was defined by the first of two consecutive time points to have a $Ca^{2+}$ ratio $\geq$ 0.2 above background. The end of a signaling event was defined by when the $Ca^{2+}$ ratio returned to <0.1 above background for two consecutive time points. Only events that had a defined beginning and end were included in calculating signal duration. For tracks with multiple events, only the first signaling event was used to calculating signal duration. For calculating speed over time, we averaged speed values from Imaris over two time points (20 s). The speed of signaling and nonsignaling time points was measured by averaging the instantaneous speed for each time point. The frequency of events was calculated by dividing the number of events compiled from all runs by the cumulative track imaging time (the sum of all of the track durations for all compiled runs), for each condition. For this calculation, events were defined as $\geq$2 consecutive time points of high $Ca^{2+}$ ratio bound by periods $\geq$ 4 min of nonsignaling.

## RNA preparation, sequencing, and analysis

Thymocytes from OT-1 Rag2$^{-/-}$, F5 Rag1$^{-/-}$, and TG6 H2$^{b/d}$ mice were sorted using a BD Influx or BD FACSAria Fusion into CD4+CD8+CXCR4+CCR7− (early positive selection), CD4+CD8+CXCR4-CCR7+ (late positive selection), and CD4−CD8+CXCR4−CCR7+ (Mature CD8SP) populations for RNA sequencing. RNA was harvested using a Quick-RNA Microprep Kit from Zymo Research (Cat. No. R1050) according to the manufacturer's instructions. RNA integrity was confirmed via Bioanalyzer and Qbit. RNA was sent to BGI Genomics for library generation and RNA sequencing. RNA was sequenced on an Illumina-HiSeq2500/4000 to a depth of 20 million reads.

Sequencing reads were processed with Trimmomatic (*Bolger et al., 2014*) to remove adapter sequences and trim low-quality bases. Reads were aligned to the mm10 genome using Bowtie 2 (*Langmead and Salzberg, 2012*), and transcripts were quantified using RSEM (*Li and Dewey, 2011*). Differential expression testing (OT-1 vs TG6, OT-1 vs F5, and F5 vs TG6 at each stage of development) was performed using DESeq2 (*Love et al., 2014*), which produced adjusted p-values corrected by the Benjamini–Hochberg procedure. Gene set enrichment analysis (GSEA) was performed for each DESeq2 comparison using FGSEA (*Korotkevich and Sukhov, 2019*), with gene sets downloaded from MSigDB (*Liberzon et al., 2011*), including the C5 collection (Gene Ontology [GO] gene sets). Following GSEA, we identified gene sets that were enriched in the order of self-reactivity (OT-1>F5>TG6 or TG6>F5>OT-1) by normalized enrichment score. Normalized counts plotted in *Figure 6—figure supplement 2b* were derived by DESeq2's median of ratio method (*Love et al., 2014*).

The curated list of ion channel genes was created by first identifying ion channels previously known to be expressed in T cells (207 genes) (*Trebak and Kinet, 2019*; *Feske et al., 2015*). We narrowed this list to include only genes that were well expressed in our RNA-seq data set (>20 reads in at least one sample) and also significantly different by DEseq2 (padj<0.05) for OT-1 vs TG6 in at least one stage of development. We also included *Tmie*, an ion channel gene differentially expressed by cells with low self-reactivity. This resulted in a list of 56 genes.

## ImmGen microarray analysis

Microarray data (RMA normalized signal intensities) were downloaded from the NCBI GEO database (*Edgar et al., 2002*; *Barrett et al., 2013*) from accession number GSE15907 (*Heng et al., 2008*) and loaded with the GEOquery package (*Davis and Meltzer, 2007*). Samples were filtered to include T.DN4.Th, T.DP.Th, and T.DP69+.Th (three replicates each). Differential expression testing between cell types was performed using Limma (*Ritchie et al., 2015*). For the analysis in *Figure 6d*, genes representative of a preselection gene expression program were identified from ImmGen

microarray data based on differential expression between preselection CD69-DP compared to DN4 and CD69+DP ($\log_2$ fold change > 1), and either combined expression of >2000 (arbitrary units) (Clk3, Ets2, Glcci1, Rorc, Plxd1, Spo11, Themis) or $\log_2$ fold change of >2 (Slc6a19, Slc7a11, Tdrd5, Wdr78). Rag1 and Rag2 were excluded. ImmGen populations were relabeled in *Figure 6—figure supplement 2c*: T.DN4.Th (DN4), T.DP.Th (DP CD69-), T.DP69+.Th (DP CD69+), T.4SP24-.Th (Mature CD4SP (T)), T.8SP24-.Th (Mature CD8SP (T)), T.4Nve.Sp (Naïve CD4SP (S)), and T.8Nve.Sp (Naïve CD8SP (S)). The gene set enrichment analysis in *Figure 6—figure supplement 2d* was performed using FGSEA (*Korotkevich and Sukhov, 2019*), with gene sets downloaded from MSigDB (*Liberzon et al., 2011*), including the C5 collection (Gene Ontology [GO] gene sets).

## Matson et al. RNA-seq analysis

RNA-seq count data were downloaded from the NCBI GEO database (*Edgar et al., 2002*; *Barrett et al., 2013*) from accession number GSE151395 (*Matson et al., 2020*). The data contained four biological replicates for each of four cell types (CD4+CD5 lo, CD4+CD5 hi, CD8+CD5 lo, CD8+CD5 hi). Differential expression testing was performed using DESeq2 (*Love et al., 2014*).

## Heatmap plotting

Heatmaps (*Figure 6b,c*, *Figure 6—figure supplement 2c*) were plotted using pheatmap with hierarchical clustering of rows using default settings. Data was scaled per row. For *Figure 6b*, rows (genes) were divided into four clusters based on hierarchical clustering. Genes in *Figure 6—figure supplement 2c* are arranged by the same ordering as in *Figure 6b*.

## Statistical analysis

Statistical analysis was performed using Prism software (GraphPad) or as explained above for the RNA-seq dataset. Specific statistical tests are indicated in the figure legends. Sample sizes were chosen based on previous publications of similar studies. p-values of <0.05 were considered significant.

# Acknowledgements

We would like to thank all the members of the Robey lab for insightful conversation and scientific advice. We would also like to thank Diana Bautista, Nadia Kurd, and Derek Bangs for critical reading of the manuscript. Two-photon imaging experiments were conducted at the CRL Molecular Imaging Center, and we would like to thank Paul Herzmark, Holly Aaron, and Feather Ives for their microscopy training and assistance. This work benefitted from data assembled by the ImmGen consortium. This work was funded by NIH RO1AI064227 (EAR). LLM and ARH were supported by NIH T32AI100829, and SA was supported by a Human Frontiers Fellowship. ZS is supported by the National Science Foundation Graduate Research Fellowship.

# Additional information

### Funding

| Funder | Grant reference number | Author |
| --- | --- | --- |
| National Institutes of Health | RO1AI064227 | Ellen A Robey |
| National Institutes of Health | T32AI100829 | Laura L McIntyre<br>Ashley R Hoover |
| Human Frontier Science Program | | Silvia Ariotti |
| National Science Foundation | GRFP | Zoë Steier |

The funders had no role in study design, data collection and interpretation, or the decision to submit the work for publication.

## Author contributions
Lydia K Lutes, Conceptualization, Data curation, Formal analysis, Validation, Investigation, Visualization, Methodology, Writing - original draft, Writing - review and editing; Zoë Steier, Resources, Data curation, Software, Formal analysis, Writing - review and editing; Laura L McIntyre, Experimentation, Writing - review and editing; Shraddha Pandey, Data curation, Investigation; James Kaminski, Data curation, Software, Formal analysis; Ashley R Hoover, Experimentation; Silvia Ariotti, Data curation; Aaron Streets, Nir Yosef, Resources, Software, Funding acquisition; Ellen A Robey, Conceptualization, Data curation, Funding acquisition, Writing - original draft, Writing - review and editing

## Author ORCIDs
Lydia K Lutes https://orcid.org/0000-0001-6133-6294
Nir Yosef http://orcid.org/0000-0001-9004-1225
Ellen A Robey https://orcid.org/0000-0002-3630-5266

## Ethics
Animal experimentation: All mice were bred and maintained under pathogen-free conditions in an American Association of Laboratory Animal Care-approved facility at the University of California, Berkeley. The University of California, Berkeley Animal Use and Care Committee approved all procedures (Animal Care and Use Protocol #AUP-2016-07-9006).

## Decision letter and Author response
Decision letter https://doi.org/10.7554/eLife.65435.sa1
Author response https://doi.org/10.7554/eLife.65435.sa2

# Additional files
## Supplementary files
• Supplementary file 1. TG6 thymocytes spend less time signaling compared to OT-1 cells. Percent time signaling was calculated by dividing the number of signaling time points by total time points.

• Supplementary file 2. TG6 thymocytes experience less frequent TCR signaling than OT-1 thymocytes. The frequency of signaling events was calculated by dividing the total number of signaling events by the cumulative track imaging time (the sum of all of the track durations for all runs) for each condition.

• Supplementary file 3. Genes upregulated in CD5lo compared to CD5hi naïve polyclonal CD8SP T cells by microarray analysis. The top 47 significantly different (p<0.05, >1.45-fold difference) genes upregulated in sorted CD5lo naive polyclonal CD8SP T cells, by microarray analysis, and ranked by their expression (fold) difference. Gene that appear multiple times represent multiple probe sets for the same gene, and a number in the left-most column is included only for the first listing. Genes of interest, Kcna2 and Tmie, are in bold. Data was analyzed by Fulton et al.(GSE62142).

• Supplementary file 4. Genes upregulated in CD5lo compared to CD5hi naïve polyclonal CD8SP T cells by RNA-seq analysis. The top 51 significantly different (padj<0.05) upregulated genes in sorted CD5lo naive polyclonal CD8+T cells and ranked by their expression (fold) difference, from the Matson et al. RNA-seq dataset (GSE151395). Genes of interest, *Kcna2* and *Tmie*, are in bold. Data was analyzed by DEseq2.

• Transparent reporting form

## Data availability
RNA-seq data have been deposited in NCBI's Gene Expression Omnibus and are accessible through GEO Series accession number GSE164896. The analysis code used for the RNA-seq data in this manuscript is available at https://github.com/YosefLab/TcellSelectionTiming. This repository also includes DE results, GSEA results, and the metadata associated with this study.

The following dataset was generated:

| Author(s) | Year | Dataset title | Dataset URL | Database and Identifier |
|---|---|---|---|---|
| Lutes LK, Steier Z, McIntyre LL, Pandey S, Kaminski J, Hoover AR, Ariotti S, Streets A, Yosef N, Robey EA | 2021 | T cell self-reactivity during thymic development dictates the timing of positive selection | https://www.ncbi.nlm.nih.gov/geo/query/acc.cgi?acc=GSE164896 | NCBI Gene Expression Omnibus, GSE164896 |

The following previously published datasets were used:

| Author(s) | Year | Dataset title | Dataset URL | Database and Identifier |
|---|---|---|---|---|
| Jameson SC, Fulton RB | 2014 | Comparison of mouse naïve (CD44lo) CD8+ T cells sorted into the highest and lowest 20% with respect to CD5 | https://www.ncbi.nlm.nih.gov/geo/query/acc.cgi?acc=GSE62142 | NCBI Gene Expression Omnibus, GSE62142 |
| Matson CA, Choi S, Zhao B, Ferenc L, Mitra AK, Love PE, Singh NJ | 2020 | CD5 dynamically calibrates basal NF-КB signaling in T cells during thymic development and peripheral activation | https://www.ncbi.nlm.nih.gov/geo/query/acc.cgi?acc=GSE151395 | NCBI Gene Expression Omnibus, GSE151395 |
| Heng TSP, Painter MW, The Immunological Genome Project Consortium | 2009 | ImmGen Microarray Phase 1 | https://www.ncbi.nlm.nih.gov/geo/query/acc.cgi?acc=GSE15907 | NCBI Gene Expression Omnibus, GSE15907 |

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

# Appendix 1

**Appendix 1—key resources table**

| Reagent type (species) or resource | Designation | Source or reference | Identifiers | Additional information |
|---|---|---|---|---|
| Genetic reagent (*M. musculus*) | C57BL/6J background | Jackson Laboratory | Stock No. 000664 RRID: IMSR_JAX: 000664 | |
| Genetic reagent (*M. musculus*) | B6.C-H2$^d$/bByJ | Jackson Laboratory | Stock No. 000359 RRID: IMSR_JAX: 000359 | |
| Genetic reagent (*M. musculus*) | B6(Cg)-*Rag2*$^{tm1.1Cgn}$/J | Jackson Laboratory | Stock No. 008449 RRID: IMSR_JAX: 008449 | |
| Genetic reagent (*M. musculus*) | B6.129S7-*Rag1*$^{tm1Mom}$ | Jackson Laboratory | Stock No. 002216 RRID: IMSR_JAX: 002216 | |
| Genetic reagent (*M. musculus*) | B6.129P2-*b2m*$^{tm1Unc}$/DcrJ | Jackson Laboratory | Stock No. 002087 RRID: IMSR_JAX: 002087 | |
| Genetic reagent (*M. musculus*) | B6.129S6-*Rag2*$^{tm1Fwa}$ Tg (TcraTcrb)1100Mjb | Taconic | Model No. 2334 RRID: IMSR_TAC: 2334 | |
| Genetic reagent (*M. musculus*) | F5 *Rag1*$^{-/-}$ | *Au-Yeung et al., 2014* | | Generated by crossing in *Au-Yeung et al., 2014*, and maintained in house |
| Genetic reagent (*M. musculus*) | B6xB6C (*H2*$^{b/d}$) | *Chu et al., 2016* | | Generated by crossing in *Chu et al., 2016*, and maintained in house |
| Genetic reagent (*M. musculus*) | TG6 *H2*$^b$ | *Chu et al., 2016* | | Generated by crossing in *Chu et al., 2016*, and maintained in house |
| Genetic reagent (*M. musculus*) | TG6 *H2*$^{b/d}$ | This paper and *Tsitsiklis et al., 2020* | | Generated by crossing TG6tg mice from *Chu et al., 2016* with B6.C-H2$^d$/bByJ mice from Jackson Labs |
| Genetic reagent (*M. musculus*) | TG6 *H2*$^{b/d}$*Rag2*$^{-/-}$ | This paper | | Generated by crossing TG6tg H2$^{b/d}$ mice with B6(Cg)-Rag2$^{tm1.1Cgn}$/J mice from Jackson Labs |
| Genetic reagent (*M. musculus*) | TG6 *H2*$^b$*Rag2*$^{-/-}$ | This paper | | Generated by crossing TG6tg H2$^b$ mice with B6(Cg)-Rag2$^{tm1.1Cgn}$/J mice from Jackson Labs |
| Genetic reagent (*M. musculus*) | OT-1 *Rag2*$^{-/-}$*B2M*$^{-/-}$ | *Ross et al., 2014* | | Generated by crossing in *Ross et al., 2014*, and maintained in house |

*Continued on next page*

*Appendix 1—key resources table continued*

| Reagent type (species) or resource | Designation | Source or reference | Identifiers | Additional information |
|---|---|---|---|---|
| Antibody | Rat Monoclonal anti-Mouse CXCR4 (Clone L276F12) PerCP-Cy5.5 | Biolegend | Cat. No. 146510 RRID:AB_2562786 | (1:200) |
| Antibody | Rat Monoclonal anti-Mouse CXCR4 (Clone L276F12) APC | Biolegend | Cat. No. 146508 RRID:AB_2562784 | (1:200) |
| Antibody | Rat Monoclonal anti-Mouse CCR7 (Clone 4B12) PE-Cy7 | Biolegend | Cat. No. 120124 RRID:AB_2616688 | (1:200) |
| Antibody | Rat Monoclonal anti-Mouse CCR7 (Clone 4B12) PE | Biolegend | Cat. No. 120106 RRID:AB_389358 | (1:200) |
| Antibody | Rat Monoclonal anti-Mouse CD4 (Clone GK1.5) APC | Tonbo Biosciences | Cat. No. 20–0041 U100 RRID:AB_2621736 | (1:200) |
| Antibody | Rat Monoclonal anti-Mouse CD4 (Clone GK1.5) APC-Cy7 | Biolegend | Cat. No. 100414 RRID:AB_312699 | (1:200) |
| Antibody | Rat Monoclonal anti-Mouse CD4 (Clone GK1.5) Pacific Blue | Biolegend | Cat. No. 100428 RRID:AB_493647 | (1:200) |
| Antibody | Rat Monoclonal anti-Mouse CD4 (Clone RM4-5) PerCP-Cy5.5 | Tonbo Biosciences | Cat. No. 65–0042 U100 RRID:AB_2621876 | (1:200) |
| Antibody | Rat Monoclonal anti-Mouse CD4 (Clone RM4-5) PE-Cy7 | Tonbo Biosciences | Cat. No. 60–0042 U100 RRID:AB_2621829 | (1:200) |
| Antibody | Rat Monoclonal anti-Mouse CD4 (Clone RM4-5) BV605 | Biolegend | Cat. No. 100548 RRID:AB_2563054 | (1:200) |
| Antibody | Rat Monoclonal anti-Mouse CD8α (Clone 53–6.7) FITC | Biolegend | Cat. No. 100706 RRID:AB_312745 | (1:200) |
| Antibody | Rat Monoclonal anti-Mouse CD8α (Clone 53–6.7) PE-Cy7 | Tonbo Biosciences | Cat. No. 60–0081 U100 RRID:AB_2621832 | (1:200) |
| Antibody | Rat Monoclonal anti-Mouse CD8α (Clone 53–6.7) APC | Tonbo Biosciences | Cat. No. 20–0081 U100 RRID:AB_2621550 | (1:200) |

*Continued on next page*

*Appendix 1—key resources table continued*

| Reagent type (species) or resource | Designation | Source or reference | Identifiers | Additional information |
|---|---|---|---|---|
| Antibody | Rat Monoclonal anti-Mouse CD8α (Clone 53–6.7) BV421 | Biolegend | Cat. No. 100738 RRID:AB_11204079 | (1:200) |
| Antibody | Rat Monoclonal anti-Mouse CD8α (Clone 53–6.7) BV605 | Biolegend | Cat. No. 100744 RRID:AB_2562609 | (1:200) |
| Antibody | Rat Monoclonal anti-Mouse CD5 (Clone 53–7.3) FITC | Biolegend | Cat. No. 100606 RRID:AB_312735 | (1:200) |
| Antibody | Rat Monoclonal anti-Mouse CD5 (Clone 53–7.3) PE | eBioscience | Cat. No. 12-0051-83 RRID:AB_465524 | (1:200) |
| Antibody | Rat Monoclonal anti-Mouse CD5 (Clone 53–7.3) PE-Cy7 | Biolegend | Cat. No. 100622 RRID:AB_2562773 | (1:200) |
| Antibody | Rat Monoclonal anti-Mouse CD5 (Clone 53–7.3) BV421 | BD Biosciences | Cat. No. 562739 RRID:AB_2737758 | (1:200) |
| Antibody | Armenian Hamster Monoclonal anti-Mouse CD69 (Clone H1.2F3) FITC | Invitrogen | Cat. No. 11-0691-85 RRID:AB_465120 | (1:200) |
| Antibody | Armenian Hamster Monoclonal anti-Mouse CD69 (Clone H1.2F3) PerCP-Cy5.5 | Biolegend | Cat. No. 104522 RRID:AB_2260065 | (1:200) |
| Antibody | Armenian Hamster Monoclonal anti-Mouse CD69 (Clone H1.2F3) PE-Cy7 | eBioscience | Cat. No. 25-0691-82 RRID:AB_469637 | (1:200) |
| Antibody | Armenian Hamster Monoclonal anti-Mouse CD69 (Clone H1.2F3) Pacific Blue | Biolegend | Cat. No. 104524 RRID:AB_2074979 | (1:200) |
| Antibody | Rat Monoclonal anti-Mouse CD24 (Clone M1/69) Pacific Blue | Biolegend | Cat. No. 101820 RRID:AB_572011 | (1:200) |
| Antibody | Armenian Hamster Monoclonal anti-Mouse TCR beta (Clone H57-597) PerCP-Cy5.5 | eBioscience | Cat. No. 45-5961-82 RRID:AB_925763 | (1:200) |
| Antibody | Armenian Hamster Monoclonal anti-mouse TCR beta (Clone H57-597) AF647 | Biolegend | Cat. No. 109218 RRID:AB_493346 | (1:200) |
| Antibody | Rat Monoclonal anti-Mouse Valpha2 (Clone B20.1) APC | Biolegend | Cat. No. 127810 RRID:AB_1089250 | (1:200) |

*Appendix 1—key resources table continued*

| Reagent type (species) or resource | Designation | Source or reference | Identifiers | Additional information |
|---|---|---|---|---|
| Antibody | Rat Monoclonal anti-Mouse Vbeta8.1/8.2 (Clone KJ16-133.18) FITC | Biolegend | Cat. No. 118406 RRID:AB_1227786 | (1:200) |
| Antibody | Rat Monoclonal anti-Mouse Vbeta2 (Clone B20.6) PE | Biolegend | Cat. No. 127908 RRID:AB_1227784 | (1:200) |
| Antibody | Mouse Monoclonal anti-Mouse Nur77 (Clone 12.14) | ThermoFisher Scientific | Cat. No. 12-5965-82 RRID:AB_1257209 | (1:100) |
| Chemical Compound, drug | 5-ethynyl-2′-deoxyuridine (EdU) | ThermoFisher Scientific | Cat. No. A10044 | |
| Chemical Compound, drug | GTG-NuSieve Agarose | Lonza | Cat. No.: 50081 | 4% in HBSS for Thymic Slicing |
| Commercial Assay or Kit | Transcription Factor Staining Buffer Set (Invitrogen) | ThermoFisher Scientific | Cat. No. 00-5523-00 | |
| Commercial Assay or Kit | Click-iT EdU Alexa Fluor 488 Flow Cytometry Assay Kit (Invitrogen) | ThermoFisher Scientific | Cat. No. C10420 | |
| Commercial Assay or Kit | Quick-RNA Microprep Kit | Zymo Research | Cat. No. R1050 | |
| Software, algorithm | GraphPad PRISM | GraphPad Software | RRID:SCR_002798 | |
| Software, algorithm | Fiji/Image J software | Fiji-Image J | https://imagej.nih.gov/ij/ RRID:SCR_003070 | |
| Software, algorithm | FlowJo | FlowJo | https://www.flowjo.com/ RRID:SCR_008520 | |
| Software, algorithm | Zen | Zeiss | RRID:SCR_013672 | |
| Software, algorithm | Imaris | Bitplane Scientific | RRID:SCR_007370 | |
| Software, algorithm | DISCit | *Moreau et al., 2012* | | |
| Software, algorithm | R Project for Statistical Computing | R Project for Statistical Computing | RRID:SCR_001905 | |
| Software, algorithm | Trimmomatic | *Bolger et al., 2014* | | |
| Software, algorithm | Bowtie 2 | *Langmead and Salzberg, 2012* | | |
| Software, algorithm | RSEM | *Li and Dewey, 2011* | | |

*Continued on next page*

*Appendix 1—key resources table continued*

| Reagent type (species) or resource | Designation | Source or reference | Identifiers | Additional information |
|---|---|---|---|---|
| Software, algorithm | DESeq2 | *Love et al., 2014* | | |
| Software, algorithm | FGSEA | *Korotkevich and Sukhov, 2019* | | |
| Software, algorithm | MSigDB | *Liberzon et al., 2011* | | |
| Software, algorithm | GEOquery | *Davis and Meltzer, 2007* | | |
| Software, algorithm | Limma | *Ritchie et al., 2015* | | |
| Software, algorithm | Ilustrator CS5 | Adobe | RRID:SCR_010279 | |
| Other | 0.4 mm transwell inserts | Corning | Cat. No. 353090 | |
| Other | Ghost Dye Violet 510 Live/Dead dye | Tonbo Biosciences | Cat. No. 13–0870 T100 | 1:1000 flow cytometry |
| Other | Cell Proliferation Dye eFluor450 | ThermoFisher Scientific | Cat. No. 65-0863-18 | 0.5 µM at 107 cells/ml at 37˚C for 15 min in PBS |
| Other | Cell Proliferation Dye eFluor670 | ThermoFisher Scientific | Cat. No. 65-0840-85 | 0.5 µM 107 cells/ml at 37˚C for 15 min in PBS |
| Other | Indo-1LR | ThermoFisher Scientific | Cat. No. I1226 | 2 mM at 3× 106 cells/ml in complete RPMI for 90 min at 37˚C. Followed by 60 min recovery in RPMI at 37˚C |

