## [Decision Letter]

**Acceptance summary:**

This study is of interest to immunologist as it fills a key knowledge gap in understanding factors involved in determining the duration of intrathymic positive selection of T cells. The findings come from a series of both in vitro and in vivo experiments implicating self-reactivity by T cell receptors on thymocytes in determining the time to completion of positive selection. An RNA-sequencing analysis suggests that gene expression differences from the pre-selection to the single-positive thymocyte stage is self-reactivity dependent, correlating in particular the level of ion channel expression with positive selection completion rates.

**Decision letter after peer review:**

Thank you for submitting your article "T cell self-reactivity during thymic development dictates the timing of positive selection" for consideration by *eLife*. Your article has been reviewed by 3 peer reviewers, one of whom is a member of our Board of Reviewing Editors, and the evaluation has been overseen by Tadatsugu Taniguchi as the Senior Editor. The following individuals involved in review of your submission have agreed to reveal their identity: JC Zúñiga-Pflücker (Reviewer #1); Nicholas Gascoigne (Reviewer #3).

Essential revisions:

1. Please address the shared concern regarding the potential role of ion channel expression by extending the discussion of their potential as highlighted in the reviewers comments.

2. The detailed comments (1-4) from reviewer #2 provide the a clear set of required revisions.

*Reviewer #1 (Recommendations for the authors):*

1. Suppl. Fig. 4 provides a wonderful analysis of the different TCR-tg DP cells undergoing selection on the thymic slices, and clearly illustrates different progression phenotypes. It is curious to see that Tg6 thymocytes appear to closely follow the Kinetic Signaling Model of selection quite nicely, while F5 and OT1 cells show an abridged version. Given the context of the present work, it would be important for the authors to discuss these results (and others in their work) in the framework of the kinetic signaling model, which would be highly beneficial to field. The need for further discussion can also extend to the recently published scRNAseq and scATACseq analysis of developing CD4+CD8+ cells adopting CD4 or CD8 lineage outcomes (Chopp et al., PMID: 33242395).

2. In Figure 1, it is curious that F5 TCR-tg cells show increased CD5 but not Nur77 expression levels compared to Tg6 cells, while OT1 are clearly much higher for both. One thought, which the authors may or may not want to discuss is the possibility that Tg6 TCRs may require β5T (Psmb11) dependent peptides for selection, like was shown for F5 TCRs (Nitta et al., PMID: 20045355), while OT1 TCRs appear to be less dependent on those peptides, which would differentially affect thymic vs peripheral peptide repertoires and thus their TCR reactivities. Perhaps, one critical difference in the selection events described by the authors is not whether the TCRs are low or high self-reactive but rather whether they require β5t processing for the peptides that then support their positive selection. If the authors agree with this possibility, then it could be mentioned in the discussion.

*Reviewer #2 (Recommendations for the authors):*

1. For Figure 2, a better use/analysis making use of the single-cell resolution of the datasets would greatly strengthen the conclusions. This would include:• An example of the raw video data (eg. a single cell before and during Ca flux) to illustrate the thresholding imposed.

• From the tracks in (b) it appears that most TG6 tracks have few Ca flux events, and the tracks that have such signaling events often have multiple subsequent events. Could this be commented on? Is it due to cells staying on the antigen presenting cell for the duration of the video, are cells circling back or are there cell-to-cell variations even within the transgenics that would be important to consider?

• For (c) a per-track summary of the data would be more informative and interpretable than the per-video summary provided. For instance: what % of tracks had any Ca events in the TG6 vs. OT1? What % of tracks had >1 flux event? It is difficult to interpret the values provided in (c) and the individual-cell resolution is lost.

2. The authors might consider merging Figures 2 and 3, such that together, these data presented then more directly address the question of frequency of interactions versus strength of signal per interaction, which is currently not clearly distinguished by the authors and is a clear advantage of their single-cell resolution measures. For the data presented in figure 3 some additional comments:

• One cause for concern is that there appears to be a clear difference in speeds of non-signaling TG6 and OT1 cells at 6 hours – why would this be?

• A more informative measure than average track speed would be the time stopped (as also shown in panels a and b), and perhaps a more direct relation (eg. x,y plot) of Ca flux duration by time spent stopped with a distinct coding of the TG6 versus OT1 cells in a single plot would make a much more compelling case for per-signal strength differences observed across transgenics.

3. The use of in vivo EdU-labelling approach (Figure 6) provides nicely complementary data to the thymic slice data in Figure 4. To strengthen the conclusions in (d-e), could the authors also provide the CD5 gMFI data for the individual transgenics? If their conclusions are correct, CD5 levels within transgenics over time in the SP population would remain constant (in contrast to the polyclonal SP). Moreover, a summary statistic of the data in (c), such as the time to half maximum SP appearance of each TCR Tg would enable statistical comparison between transgenics. Finally, why is the CD5 decrease is less striking in the CD8 SP compared to the CD4 SP population (panel f)?

4. The summary figure 8 is confusing, introduces new terminology not previously used (phase 1 versus phase 2 genes), and includes a graph on 'TCR sensitivity over time' which is not based on data provided in this study. Please simplify to summarize the model based on the data in the paper (perhaps a diagram tracking a low versus high self-reactivity cell would be more useful here, highlighting stage transitions and accompanying changes in gene expression).

Reviewer #3 (Recommendations for the authors):

None.

---

## [Author Response]

Reviewer #1 (Recommendations for the authors):1. Suppl. Fig. 4 provides a wonderful analysis of the different TCR-tg DP cells undergoing selection on the thymic slices, and clearly illustrates different progression phenotypes. It is curious to see that Tg6 thymocytes appear to closely follow the Kinetic Signaling Model of selection quite nicely, while F5 and OT1 cells show an abridged version. Given the context of the present work, it would be important for the authors to discuss these results (and others in their work) in the framework of the kinetic signaling model, which would be highly beneficial to field. The need for further discussion can also extend to the recently published scRNAseq and scATACseq analysis of developing CD4+CD8+ cells adopting CD4 or CD8 lineage outcomes (Chopp et al., PMID: 33242395).

The reviewer astutely notes that a CD4+CD8lo population is detectable amongst TG6 thymocytes developing in the thymic slice culture, but is less prominent in the F5 and OT1 models, and relates this observation to the frequently cited “Kinetic Signaling” model for CD4 vs CD8 development. Interestingly, intact OT1 TCR transgenic mice do contain a prominent CD4+CD8lo population, although this population is much less prominent when preselection OT1 thymocytes give rise to CD8 SP thymocytes in thymic slice culture (Ross et al., 2014, PMID: 24927565 [Fig 1]; and current study). It is also noteworthy that while some MHC-1 TCR transgenic models have a prominent CD4+CD8lo population (OT1, P14), others do not (HY, F5, TG6). The observation that the CD8 SP thymocyte can efficiently develop under conditions in which they do not pass through an obvious CD4+CD8lo stage suggests that this may not be an obligatory developmental intermediate for CD8 T cell development. This is also in line with early observations that constitutive expression of CD8 transgenes, which also prevent the appearance of a CD4+CD8lo population, does not prevent the efficient development of CD8 SP (Robey et al., 1991, PMID: 1898873; and numerous subsequent studies). Given that it would require a lengthy section of text to adequately discuss this issue and given there are already numerous published models for CD8 development that do not align perfectly with the kinetic signaling model, we would prefer to keep the focus of the discussion on the impact of self-reactivity within MHC-1 specific thymocytes destined for the CD8 lineage.

2. In Figure 1, it is curious that F5 TCR-tg cells show increased CD5 but not Nur77 expression levels compared to Tg6 cells, while OT1 are clearly much higher for both. One thought, which the authors may or may not want to discuss is the possibility that Tg6 TCRs may require β5T (Psmb11) dependent peptides for selection, like was shown for F5 TCRs (Nitta et al., PMID: 20045355), while OT1 TCRs appear to be less dependent on those peptides, which would differentially affect thymic vs peripheral peptide repertoires and thus their TCR reactivities. Perhaps, one critical difference in the selection events described by the authors is not whether the TCRs are low or high self-reactive but rather whether they require β5t processing for the peptides that then support their positive selection. If the authors agree with this possibility, then it could be mentioned in the discussion.

The reviewer raises an intriguing point regarding whether self-reactivity may be related to the dependence of the TCRs on β5T for their development. Indeed the β5T dependence of the TCR transgenic models shown in Nitta et al. shows an overall inverse correlation with CD5 expression; with F5 (CD5 intermediate) being relatively dependent, and OT-1 (CD5 high) independent of β5T. However, for the set of 5 TCR examined in that study, the correlation between CD5 and β5T dependence was not perfect. Interestingly, our data in Figure 1 show a decrease in CD5 and Nur77 expression when comparing F5 lymph node T cells to thymic CD8 SP, suggesting a drop in F5 self-reactivity from the thymus to the periphery due to a loss of β5T-derived peptides in the periphery. In contrast, TG6 maintains similar or slightly higher levels of CD5 and Nur77 in the periphery, suggesting it may not preferentially recognize β5T-derived peptides. Because we have no information regarding the β5T dependence of TG6, we have chosen not to discuss this topic in detail in the manuscript. However, we have updated the text to acknowledge the impact β5T may have on shifting CD5 and Nur77 expression of F5.

Reviewer #2 (Recommendations for the authors):1. For Figure 2, a better use/analysis making use of the single-cell resolution of the datasets would greatly strengthen the conclusions. This would include:• An example of the raw video data (eg. a single cell before and during Ca flux) to illustrate the thresholding imposed.

We have modified our supplemental videos to show the raw data, the ratiometric image, and the graph indicated calcium ratio and speed over time for each cell shown.

• From the tracks in (b) it appears that most TG6 tracks have few Ca flux events, and the tracks that have such signaling events often have multiple subsequent events. Could this be commented on? Is it due to cells staying on the antigen presenting cell for the duration of the video, are cells circling back or are there cell-to-cell variations even within the transgenics that would be important to consider?

While we occasionally observe individual cells undergoing multiple signaling events there are also many examples of tracks with single signaling events, and we have not noted any difference between TG6 versus OT-1 in the prevalence of multiple signaling events. In addition, the signaling cells that experience subsequent events are not arrested in one location but remain motile between events. Therefore, we consider it unlikely that they are remaining in close proximity to the same APC.

• For (c) a per-track summary of the data would be more informative and interpretable than the per-video summary provided. For instance: what % of tracks had any Ca events in the TG6 vs. OT1? What % of tracks had >1 flux event? It is difficult to interpret the values provided in (c) and the individual-cell resolution is lost.

We now include a per track summary, as well as a per run summary, of the data in Figure 2—figure supplement 1.

2. The authors might consider merging Figures 2 and 3, such that together, these data presented then more directly address the question of frequency of interactions versus strength of signal per interaction, which is currently not clearly distinguished by the authors and is a clear advantage of their single-cell resolution measures.

We have combined Figures 2 and 3 (new Figure 2) in the revised manuscript..

For the data presented in figure 3 some additional comments:• One cause for concern is that there appears to be a clear difference in speeds of non-signaling TG6 and OT1 cells at 6 hours – why would this be?• A more informative measure than average track speed would be the time stopped (as also shown in panels a and b), and perhaps a more direct relation (eg. x,y plot) of Ca flux duration by time spent stopped with a distinct coding of the TG6 versus OT1 cells in a single plot would make a much more compelling case for per-signal strength differences observed across transgenics.

Thymocytes reduce their speed, but do not stop, during signaling events. Thus, the comparison of the average speed during signaling and nonsignaling portions of the tracks shown in Figure 3c (Figure 2e and f of the revised manuscript) accurately captures the differences in speed that relate to signaling during positive selection.

The reviewer also astutely notes that at 6 hours, during non-signaling timepoints, OT-1 cells are migrating more quickly than TG6 cells. There is indeed a significant difference in the speed of the nonsignaling portions of track for OT-1 and TG6 thymocytes (Author response image 1). We reported previously that thymocytes increase their overall speed as they progress through positive selection (Ross et al., 2014, PMID: 24927565, Figure 2a and b). Thus the increase in basal (nonsignaling) speed for OT-1 compared to TG6 thymocytes after 6 hours of culture in a positive selecting environment is a reflection of their more rapid maturation. When we calculate the signal-associated pausing, we compare the speed within the same track during the signaling versus nonsignaling portions of the track to adjust for this difference.

**Author response image 1. sa2fig1:** Average speed of nonsignaling (nonsig) timepoints in tracks with at least one signaling event. Each dot represents a single track. Data are presented as average ± SD and analyzed using a Student’s T-test (****P<0.0001). All data are compiled from two or more experiments.

3. The use of in vivo EdU-labelling approach (Figure 6) provides nicely complementary data to the thymic slice data in Figure 4. To strengthen the conclusions in (d-e), could the authors also provide the CD5 gMFI data for the individual transgenics? If their conclusions are correct, CD5 levels within transgenics over time in the SP population would remain constant (in contrast to the polyclonal SP).

We appreciate this suggestion, and the requested data has been included in a new supplemental figure, Figure 5—figure supplement 1 in the revised manuscript. The EdU+ CD8SP CD5 gMFI is not significantly different within each transgenic regardless of the day post injection.

Moreover, a summary statistic of the data in (c), such as the time to half maximum SP appearance of each TCR Tg would enable statistical comparison between transgenics.

Again, we appreciate the reviewer’s excellent suggestion. In the revised version of the manuscript, we have added Figure 5c (right panel). This panel displays the percent maximal EdU CD8SP accumulation for each transgenic, at each timepoint post EdU injection. Additionally, we provide statistical analysis for this figure.

Finally, why is the CD5 decrease is less striking in the CD8 SP compared to the CD4 SP population (panel f)?

CD4SP cells have higher CD5 in general and emerge more quickly compared to

CD8SP thymocytes (Azzam et al., 1998; Saini et al. 2010; Lucas, Vasseur, and Penit 1993). Thus, at day 4 post EdU injection we are able to capture a larger population of EdU+ CD4SP cells compared to CD8SPs, which could result in the significant difference we see in CD5 expression between day 4 and day 7 EdU+ CD4s.

4. The summary figure 8 is confusing, introduces new terminology not previously used (phase 1 versus phase 2 genes), and includes a graph on 'TCR sensitivity over time' which is not based on data provided in this study. Please simplify to summarize the model based on the data in the paper (perhaps a diagram tracking a low versus high self-reactivity cell would be more useful here, highlighting stage transitions and accompanying changes in gene expression).

We thank the reviewer for the suggestions, and we have modified the figure to improve the clarity. We have also edited the text in the discussion to temper our concluding statements regarding ion channels, especially those relating to TCR.